# The mechanosensitive ion channel PIEZO1 promotes satellite cell function in muscle regeneration

Kotaro Hirano[1,2], Masaki Tsuchiya[1,3] , Akifumi Shiomi[4], Seiji Takabayashi[1], Miki Suzuki[2], Yudai Ishikawa[2], Yuya Kawano[2], Yutaka Takabayashi[2], Kaori Nishikawa[4], Kohjiro Nagao[1], Eiji Umemoto[2], Yasuo Kitajima[5] , Yusuke Ono[6] , Keiko Nonomura[7,8,9], Hirofumi Shintaku[4] , Yasuo Mori[1], Masato Umeda[1], Yuji Hara[2] 

**Muscle satellite cells (MuSCs), myogenic stem cells in skeletal muscles, play an essential role in muscle regeneration. After skeletal muscle injury, quiescent MuSCs are activated to enter the cell cycle and proliferate, thereby initiating regeneration; however, the mechanisms that ensure successful MuSC division, including chromosome segregation, remain unclear. Here, we show that PIEZO1, a calcium ion ($Ca^{2+}$)-permeable cation channel activated by membrane tension, mediates spontaneous $Ca^{2+}$ influx to control the regenerative function of MuSCs. Our genetic engineering approach in mice revealed that PIEZO1 is functionally expressed in MuSCs and that *Piezo1* deletion in these cells delays myofibre regeneration after injury. These results are, at least in part, due to a mitotic defect in MuSCs. Mechanistically, this phenotype is caused by impaired PIEZO1-Rho signalling during myogenesis. Thus, we provide the first concrete evidence that PIEZO1, a bona fide mechanosensitive ion channel, promotes proliferation and regenerative functions of MuSCs through precise control of cell division.**

## Introduction

Muscle-resident stem cells, called muscle satellite cells (MuSCs), are critical for skeletal muscle regeneration after muscle injury. Under resting conditions, quiescent MuSCs reside on the plasma membrane of myofibres and underneath the extracellular matrix (Mauro, 1961). Upon injury, they get activated and undergo differentiation to become fusogenic myoblasts that repair damaged myofibres or form multinucleated cells called myotubes (Relaix & Zammit, 2012). The importance of MuSCs is further demonstrated by the fact that their impaired function is closely associated with

sarcopenia and a class of muscle diseases (i.e., muscular dystrophy) (Dumont et al, 2015; Tierney & Sacco, 2016; Feige et al, 2018).

An important characteristic of MuSCs is that a small population of these cells can give rise to a large number of myofibres in engrafted muscle. Moreover, MuSCs retain the capacity to self-renew (Collins et al, 2005). Extensive efforts have been made to elucidate the mechanisms underlying MuSC activation, proliferation, and differentiation. MuSCs express paired box 7 (Pax7), a transcription factor essential for MuSC-specific gene expression (Seale et al, 2000). Once activated, the expression of distinct transcription factors, including Myf5 and MyoD, is up-regulated, generating myoblast-specific gene signatures (Relaix & Zammit, 2012). The fate of MuSCs is determined by several signalling cascades and multiple secretory molecules (Rodgers et al, 2017; Eliazer et al, 2019). In addition to biochemical cues, biophysical stimuli are also thought to influence the functions of MuSCs (Evano & Tajbakhsh, 2018; Li et al, 2018). Although the molecular signalling pathways involved in MuSC function have been identified, the critical determinant that integrates the changes in mechanical properties upon myofibre injury into biological processes that stimulate MuSCs remains to be elucidated.

The cytosolic concentration of $Ca^{2+}$ is strictly maintained within the nanomolar range and is approximately 20,000-fold lower than that in the extracellular fluids (Clapham, 2007). Thus, $Ca^{2+}$ influx across the plasma membrane is recognised as one of the critical determinants of diverse cellular and physiological events (Berridge et al, 2003). Multiple $Ca^{2+}$ channels play critical roles in skeletal muscle function; the L-type voltage-gated $Ca^{2+}$ channel, also known as a dihydropyridine receptor, is essential for myofibre contraction via its interaction with the ryanodine receptor (Campbell et al, 1988; Calderon et al, 2014). On the other hand, the STIM1-ORAI complex, a component of store-operated $Ca^{2+}$ channel, is involved in muscle development, growth, and physiology (Endo et al, 2015; Michelucci et al, 2018). Among the known categories of $Ca^{2+}$ channels, the

[1]Department of Synthetic Chemistry and Biological Chemistry, Graduate School of Engineering, Kyoto University, Kyoto, Japan   [2]School of Pharmaceutical Sciences, University of Shizuoka, Shizuoka, Japan   [3]PRESTO, JST, Kawaguchi-shi, Saitama, Japan   [4]Microfluidics RIKEN Hakubi Research Team, RIKEN Cluster for Pioneering Research, Wako, Saitama, Japan   [5]Department of Immunology, Graduate School of Biomedical and Health Sciences, Hiroshima University, Hiroshima, Japan   [6]Department of Muscle Development and Regeneration, Institute of Molecular Embryology and Genetics, Kumamoto University, Kumamoto, Japan   [7]Division of Embryology, National Institute for Basic Biology, Aichi, Japan   [8]Department of Basic Biology, School of Life Science, SOKENDAI, Okazaki, Japan   [9]Department of Life Science and Technology, Tokyo Tech, Yokohama, Japan

Correspondence: yhara@u-shizuoka-ken.ac.jp

mechanosensitive ones that are activated by physical stimuli at the plasma membrane, are thought to be plausible candidates as regulators of MuSC functions.

PIEZO1, a mechanosensitive ion channel activated by membrane tension, plays a fundamental role in sensing biophysical forces (Coste et al, 2010). It is composed of roughly 2,500 amino acid residues that form a propeller-like homo-trimer and is well-conserved from plants to human. *PIEZO1* gene mutations have been identified in patients with hereditary xerocytosis, where it confers resistance to malaria (Ma et al, 2018). Moreover, studies on tissue-specific *Piezo1*-deficient mice have revealed that PIEZO1 is involved in the mechanosensation of cells and tissues, including neuronal progenitor cells (Pathak et al, 2014), chondrocytes (Lee et al, 2014), and blood and lymphatic vessels (Li et al, 2014; Nonomura et al, 2018), suggesting that PIEZO1 is critical for tissue homeostasis (Murthy et al, 2017). Our previous work indicated that PIEZO1 is highly expressed in myoblasts and that the ion channel activity of PIEZO1 is positively regulated by phospholipid flippase, an enzyme that catalyses the translocation of phospholipids from the outer to inner leaflets of the plasma membranes. Moreover, our results demonstrated that *Piezo1* deletion leads to impaired actomyosin assembly, causing the formation of abnormally enlarged myotubes, suggesting that PIEZO1-mediated $Ca^{2+}$ influx is a critical determinant of myotube morphogenesis (Tsuchiya et al, 2018). However, the role of PIEZO1 in MuSCs remains unclear.

In this study, we evaluated the expression of PIEZO1 in skeletal muscles and showed that PIEZO1 is highly expressed in undifferentiated MuSCs. We demonstrated that *Piezo1* deficiency in MuSCs leads to delayed myofibre regeneration after myolysis-induced degeneration. Our results using a series of genetic mouse models revealed that this phenotype resulted, at least in part, from impaired activation, proliferation, and mitosis of MuSCs caused by suppression of the Rho-mediated signalling pathway. Thus, our results indicate that PIEZO1 is involved in skeletal muscle regeneration by promoting MuSC function.

# Results

## PIEZO1 is predominantly expressed in MuSCs

We previously reported that PIEZO1 is critical for morphogenesis during myotube formation (Tsuchiya et al, 2018). To examine the expression profile of *Piezo1* in skeletal myogenesis, we isolated total RNA from freshly isolated MuSCs (FISCs), proliferating MuSCs cultured for 4 d in growth medium, and mature skeletal muscle tissue. Quantitative RT-PCR analysis indicated that *Piezo1* mRNA expression was moderate and high in FISCs and proliferating MuSCs, respectively. However, no clear *Piezo1* expression was observed in mature myofibres (Fig 1A).

To validate the protein expression of PIEZO1 in MuSCs, we used *Piezo1-tdTomato* mice in which the C-terminus of endogenous PIEZO1 was fused with a red fluorescent protein, tdTomato (Fig S1A; Ranade et al, 2014). The protein expression of PIEZO1-tdTomato in proliferating MuSCs was validated by Western blot analysis (Fig 1B). Immunofluorescent analysis was performed to detect PIEZO1-

tdTomato and Pax7, a MuSC-specific transcription factor (Seale et al, 2000), on floating single myofibres isolated from the extensor digitorum longus (EDL) muscle of *Piezo1-tdTomato* mice. PIEZO1 expression was clearly observed in Pax7-positive MuSCs but not in myofibres (Fig 1C and D), which is consistent with the RT-PCR results shown in Fig 1A. Furthermore, PIEZO1 expression was also observed in Pax7-positive FISCs isolated from *Piezo1-tdTomato* muscle samples using fluorescence-activated cell sorting (FACS) as a population of VCAM1$^{+ve}$, Sca1$^{-ve}$, CD31$^{-ve}$, and CD45$^{-ve}$ cells (Figs 1D and S1B). PIEZO1-tdTomato was detected in almost all Pax7-positive FISCs (Fig 1E).

We next examined the localization of PIEZO1-tdTomato in cultured MuSCs and myofibres and found that PIEZO1-tdTomato was localized at the plasma membrane of Pax7- or MyoD-positive MuSCs, whereas PIEZO1-tdTomato was accumulated intracellularly in myogenin-positive MuSCs (Fig S1C and D). In support of this data, PIEZO1-tdTomato was clearly observed in undifferentiated cells, whereas it was only marginally detected in myosin heavy chain-positive cells (i.e., differentiated myotubes) (Fig S1E).

MuSCs are committed to becoming myoblasts during myogenesis in response to myofibre injury (Yin et al, 2013). To investigate whether PIEZO1 is expressed in activated MuSCs, the tibialis anterior (TA) muscle was injected with cardiotoxin (CTX), a venom toxin that causes myofibre degeneration and concomitant regeneration. This was followed by the immunofluorescent detection of PIEZO1-tdTomato on regenerating myofibres 4 d after CTX injection. PIEZO1 expression was detected in the Pax7-positive cell population (Fig S1F), suggesting that PIEZO1 is involved in myogenesis.

## PIEZO1 is involved in $Ca^{2+}$ mobilisation in MuSCs

To examine whether PIEZO1 acts as a $Ca^{2+}$-permeable channel in MuSCs, we generated *Piezo1*-deficient mice. As the systemic deletion of *Piezo1* gene causes embryonic lethality (Li et al, 2014), we used conditional gene targeting using Cre-loxP-mediated genetic recombination. Mice harbouring "floxed" alleles of the *Piezo1* gene were crossed with *Pax7$^{CreERT2/+}$*, a transgenic mouse line that specifically expresses Cre-recombinase in MuSCs under the control of tamoxifen (TMX) (Fig S2A; Lepper & Fan, 2010). After the resultant mice (*Piezo1$^{flox/flox}$; Pax7$^{CreERT2/+}$*) were obtained, *Piezo1* gene deletion was induced by intraperitoneal injection of tamoxifen in the mice (called *Piezo1* cKO) for five consecutive days. MuSCs isolated from *Piezo1* cKO were seeded onto glass-bottom dishes and subjected to $Ca^{2+}$ measurements using the ratiometric $Ca^{2+}$ indicator Fura-2. $Ca^{2+}$ influx was clearly detected with Yoda-1 (a chemical agonist for PIEZO1; Syeda et al, 2015) in FISC and cultured MuSCs from controls but not from *Piezo1* cKO, confirming PIEZO1 expression and the effectiveness of *Piezo1* conditional deletion in these cells (Figs 2C and D and S2B–G).

We investigated whether $Ca^{2+}$ fluctuations (Berridge et al, 2003) could be detected in isolated MuSCs. $Ca^{2+}$ measurements clearly detected spontaneous $Ca^{2+}$ transients in HEPES-buffered saline (HBS) containing 2 mM $Ca^{2+}$ but were almost completely abolished by chelating external $Ca^{2+}$ with EGTA (Fig 2A and B). This allowed us to examine the contribution of PIEZO1 to these spontaneous $Ca^{2+}$ transients in MuSCs. Indeed, a significant reduction in $Ca^{2+}$ transients was evident in *Piezo1* cKO MuSCs (Fig 2C, E, and F). These

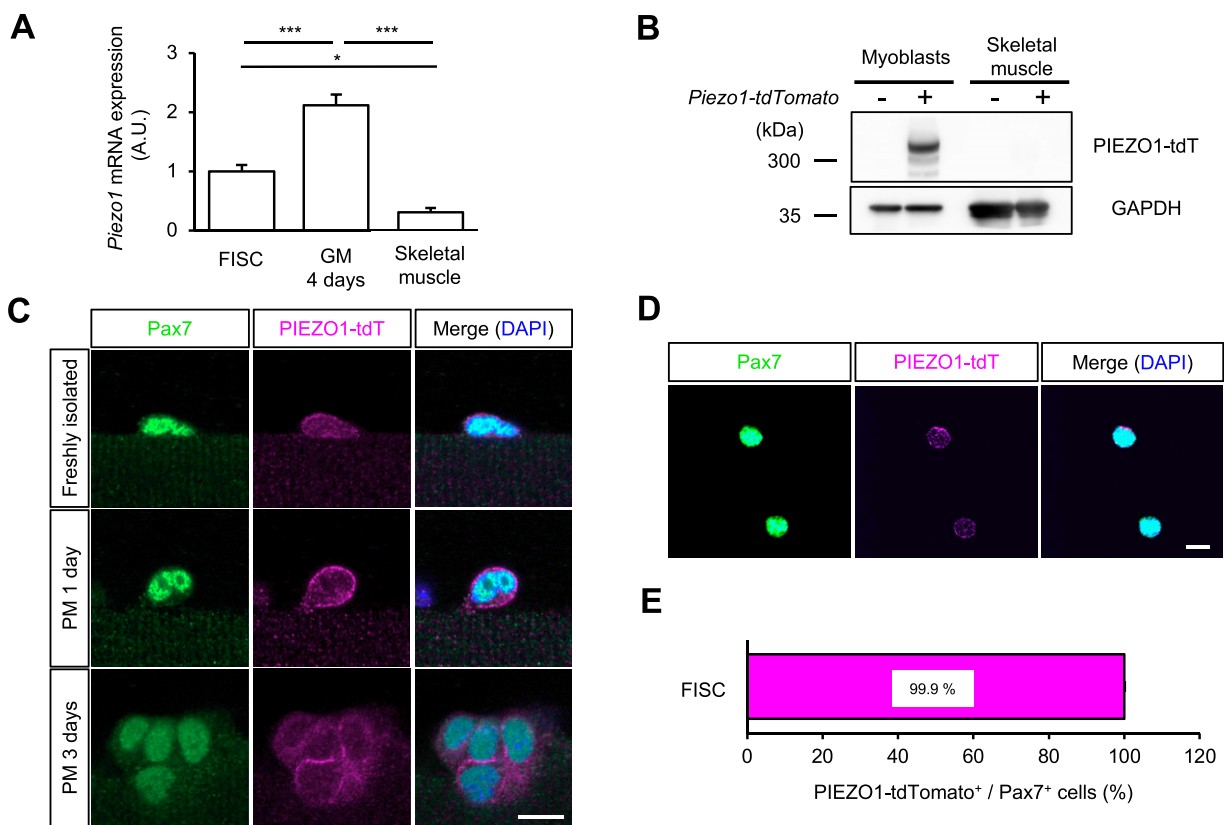

**Figure 1. Expression of PIEZO1 channel in undifferentiated muscle satellite cells (MuSCs).**
**(A)** Quantitative RT–PCR analysis of the *Piezo1* gene in MuSCs. Total RNA samples were extracted from freshly isolated satellite cells (FISCs), proliferating satellite cells (GM 4 d), and mature skeletal muscle. 18*S* ribosomal RNA expression was used as a control (N = 3 mice per condition). **(B)** Western blot analysis of PIEZO1-tdTomato (PIEZO1-tdT) in MuSCs cultured in GM for 6 d or skeletal muscle isolated from *Piezo1-tdTomato* mice or C57BL6 wild-type mice. GAPDH was used as a loading control. **(C, D)** Detection of PIEZO1 protein in MuSCs. Immunofluorescent analysis of PIEZO1 on isolated myofibres from EDL muscle (C) or FISCs (D) from *Piezo1-tdTomato* mice. Isolated myofibres were cultured for 1 or 3 d in plating medium (PM). PIEZO1-tdTomato was visualised by immunofluorescent staining with anti-RFP antibody (magenta). Pax7 and nuclei were also detected with anti-Pax7 antibody (green) and DAPI (blue), respectively. Scale bar: 10 $\mu$m. **(E)** Percentage of PIEZO1-tdTomato-positive cells in Pax7-positive FISCs (>100 cells per condition from N = 3 mice). *$P$ < 0.05, ***$P$ < 0.001. (Tukey's test).
Source data are available online for this figure.

results indicate that PIEZO1 acts as a $Ca^{2+}$-permeable ion channel that predominantly generates $Ca^{2+}$ fluctuations in MuSCs.

Next, we investigated whether $Ca^{2+}$ affects the proliferative capacity of MuSCs by chelating cytosolic $Ca^{2+}$ in MuSCs using 1,2-bis(2-aminophenoxy) ethane-*N,N,N',N'*-tetraacetic acid tetra(acetoxymethyl ester) (BAPTA-AM), a membrane-permeable $Ca^{2+}$ chelator. During a 6-h pulse of 5-ethynyl-20-deoxyuridine (EdU) with BAPTA-AM, the number of EdU-positive proliferating MuSCs decreased in BAPTA-AM-treated cells (Fig 2G–I). This indicates that $Ca^{2+}$ signalling is essential for MuSC proliferation.

### PIEZO1 plays a role in myofibre regeneration

Based on the results of immunofluorescent analyses and $Ca^{2+}$ imaging, we hypothesised that PIEZO1 plays a role in myofibre regeneration. We examined the effect of *Piezo1* deletion on MuSC function, by using CTX to induce the degeneration and subsequent regeneration of myofibres (Fig 3A). The reduction in muscle weight was more evident in *Piezo1* cKO mice than in control mice at 7 and 14 d post-CTX injection (Figs 3A–C and S3A–D).

This result led us to further evaluate the histological abnormalities during myofibre regeneration after CTX injection. Haematoxylin and eosin staining revealed myofibres with centrally located nuclei (indicating regenerating myofibres) in the wild-type muscle at 7 d post-injection with CTX. In contrast, *Piezo1* cKO muscle displayed hallmarks of myofibre regeneration defects, such as fibrosis (Fig 3D–F). These abnormalities were further confirmed by a reduction in the cross-sectional area (CSA) of regenerating myofibres at 7 d post-injection with CTX (Fig 3G and H), although there was no histological abnormality in intact muscle or at later time points (Figs 3D and S3E–I). In addition, *Piezo1*$^{LacZ/+}$, a mouse line heterologously harbouring *LacZ* and neomycin-resistant gene cassettes that disrupt *Piezo1* gene function, did not show obvious abnormalities (Fig S4A). Importantly, no obvious histological abnormalities were observed in *Piezo1*$^{+/+}$; *Pax7*$^{CreERT2/+}$ mice, eliminating the possibility that heterologous deletion in the *Pax7* gene affects the regeneration capacity of myofibres in *Piezo1* cKO mice in our study (Fig S4B and C). These results indicate that PIEZO1 is involved in myofibre regeneration after muscle injury.

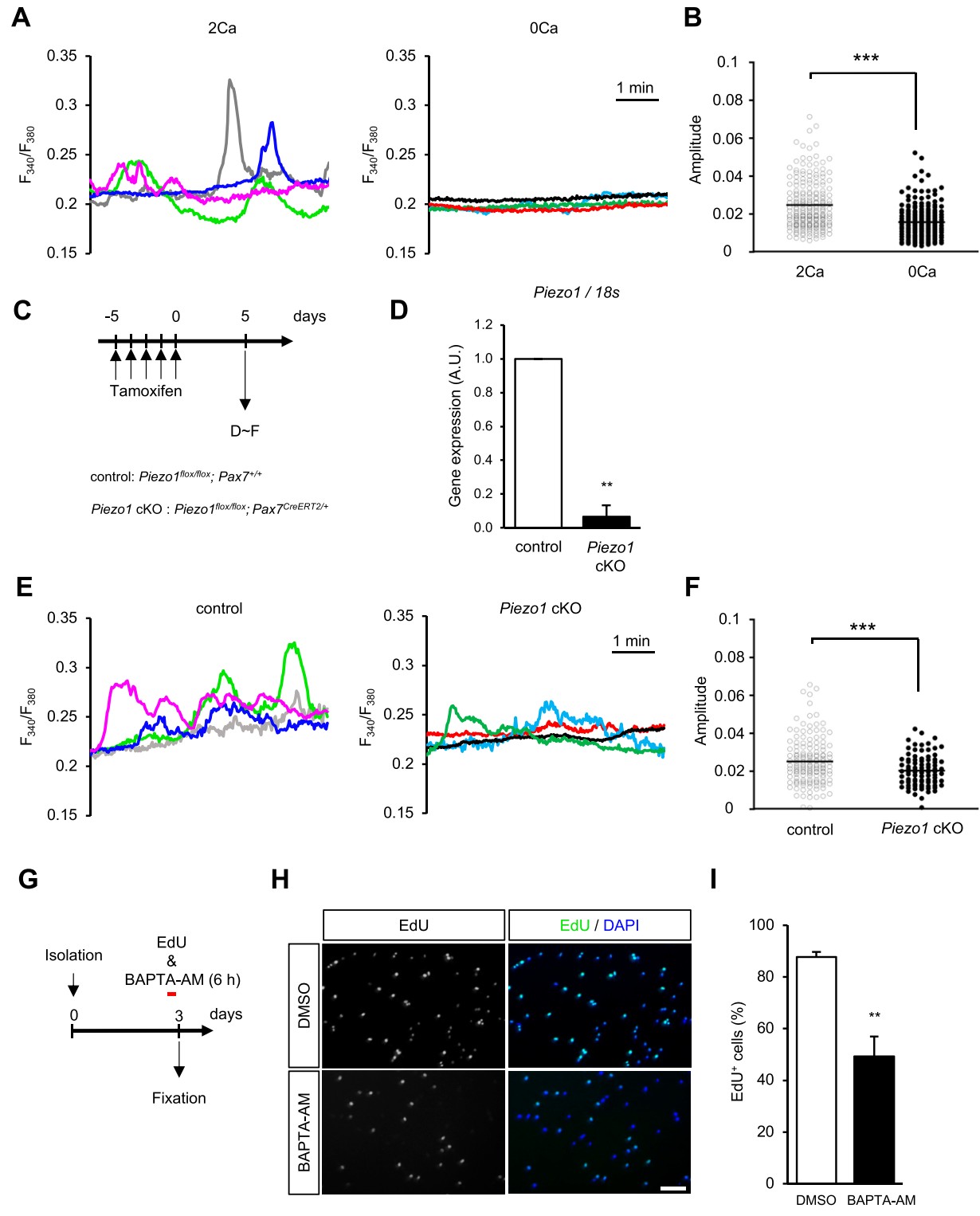

**Figure 2. PIEZO1-dependent Ca²⁺ fluctuation in muscle satellite cells (MuSCs).**

**(A, B)** Calcium ion (Ca²⁺) measurements in MuSCs under resting conditions. After isolation of MuSCs from control mice, Ca²⁺ fluctuations were monitored using the Ca²⁺ indicator Fura-2. **(A)** Representative traces of Ca²⁺ fluctuations in control MuSCs in the presence (2 mM Ca²⁺; left panel with light green, magenta, grey, and dark blue traces) and absence (0 mM Ca²⁺; right panel with dark green, red, black, and light blue traces) of extracellular Ca²⁺. **(B)** Measurement of the amplitude of Ca²⁺ fluctuations in MuSCs. Open circles: amplitudes in the presence of Ca²⁺, closed circles: those in the absence of Ca²⁺ (>130 cells per condition from N = 3 mice). **(C, D)** Induction of *Piezo1*-deficiency by administration of tamoxifen via intraperitoneal injection. **(C)** Time course for induction of *Piezo1*-deficiency. **(D)** Relative expression of *Piezo1* mRNA in freshly isolated satellite cells derived from control and *Piezo1* cKO mice after tamoxifen injection. Data represent means + SEM. **(E, F)** Calcium ion (Ca²⁺) measurements in control or *Piezo1* cKO MuSCs. **(E)** Representative traces of control and *Piezo1* cKO cells are shown in the left and right panels, respectively. **(F)** Measurement of the

## PIEZO1 regulates the activation and proliferation of MuSCs

To investigate the role of PIEZO1 in myofibre regeneration, we sought to evaluate the number of MuSCs in CTX-injected TA muscle. Immunofluorescent detection of MuSCs with an anti-Pax7 antibody revealed that the number of MuSCs on muscle sections from *Piezo1* cKO mice was significantly reduced compared with that from control mice (Fig 4A–C), at 7 d after cardiotoxin administration. This suggests that PIEZO1 plays a role in the maintenance of the MuSC pool. To examine whether the proliferation of MuSCs was affected by *Piezo1* deletion, we performed EdU incorporation assays. After CTX injection into the TA muscle, the mice were subjected to intraperitoneal injection with EdU, followed by the detection of incorporated EdU in the nuclei of MuSCs, representing active DNA synthesis at 3 d post-CTX injection (Fig 4D). As indicated in Fig 4E and F, the number of EdU-positive and M-cadherin-positive cells (i.e., MuSCs that entered the cell cycle) was clearly reduced in the muscle sections of *Piezo1* cKO compared with the control. We further examined whether this trend was observed in the isolated MuSCs. Surprisingly, the number of EdU-positive MuSCs increased significantly in *Piezo1* cKO mice at 40 h post-isolation, in clear contrast to those cultured for 3 d (Fig 4G–J). These results suggest that PIEZO1 has multiple roles in MuSCs: suppression of MuSC activation by preventing cell cycle entry at an early stage and enhancing cell proliferation at a later stage post-injury.

To further evaluate the capacity for myogenic progression, we performed immunofluorescent analysis on isolated myofibres, where the microenvironment surrounding MuSCs was relatively preserved. In the case of freshly isolated myofibres, the number of Pax7-positive MuSCs in *Piezo1* cKO was comparable to that in the control samples (Fig 5A–C). We next examined the expression levels of MyoD, a myogenic regulatory factor that is predominantly expressed in MuSCs after activation (Zammit et al, 2004). The isolated myofibres were cultured in plating medium (Figs 5A and S5A) and subjected to immunofluorescent analysis. After 30 h of culture, the proportions of Pax7-positive/MyoD-negative (i.e., self-renewed cells), Pax7-positive/MyoD-positive (i.e., activated or proliferative cells), and Pax7-negative/MyoD-positive (i.e., differentiated cells) groups (Ono et al, 2011) in *Piezo1* cKO mice were comparable to those in control mice (Fig 5D and E). However, the number of EdU-positive MuSCs was clearly increased in *Piezo1* cKO mice compared with the control (Fig 5F). This phenomenon was further confirmed by the detection of Ki67 (a proliferation marker) (Fig S5A and B), indicating that *Piezo1*-deficient MuSCs are prone to enter the cell cycle at an earlier time point.

We also examined the MuSCs when almost all of them were undergoing proliferation, i.e., 2 or 3 d post-isolation. Our results reveal that although the number of cells was significantly reduced in *Piezo1* cKO mice, the proportions of Pax7-positive/MyoD-negative (i.e., self-renewed cells), Pax7-positive/MyoD-positive (i.e., activated or proliferative cells), and Pax7-negative/MyoD-positive

(i.e., differentiated cells) groups (Ono et al, 2011) in *Piezo1* cKO mice were comparable to those in control mice (Figs 5G and H and S5A, C, and D). The same trend was observed when the myofibres were co-stained with anti-Pax7 and anti-Myogenin antibodies: the proportion of Pax7-positive/Myogenin-negative (i.e., quiescent or activated cells) and Pax7-negative/Myogenin-positive (i.e., differentiated cells) groups in *Piezo1* cKO mice was comparable to that in control mice (Fig S5A, E, and F).

To further confirm the effects of *Piezo1*-deficiency on MuSC activation and proliferation, we used *Rosa26*$^{YFP/+}$ mice, in which the YFP protein is expressed in MuSCs under the control of the Cre recombinase. We evaluated MuSCs in *Piezo1*$^{+/+}$; *Pax7*$^{CreERT2/+}$; *Rosa26*$^{YFP/+}$ mice and *Piezo1*$^{flox/flox}$; *Pax7*$^{CreERT2/+}$; *Rosa26*$^{YFP/+}$ mice (YFP mice and cKO YFP mice, respectively). In the absence of injury, the number of YFP-positive cells in cKO YFP mice was comparable to that in YFP mice (Fig S6A–C). In clear contrast, cKO YFP mice displayed a reduced number of MuSC-derived cells at 7 d post-CTX injection (Fig S6D–F). We also examined the status of MuSCs on isolated myofibres and confirmed the same phenotypes in cKO YFP mice (i.e., increased number of Ki67$^+$ or EdU$^+$ cells when cultured for 30 h, and decreased number of YFP$^+$ cells per myofibres when cultured for 3 d) (Fig S6G–M).

The features of the microenvironment, such as the matrix stiffness, are known to have a major impact on the growth and fate of the MuSCs (Gilbert et al, 2010; Quarta et al, 2016, 2017; Silver et al, 2021). To examine whether *Piezo1* deficiency affects the ability to sense changes in substrate elasticity, isolated MuSCs were cultured onto soft (2-kPa) and hard (32-kPa) substrates. Increase in stiffness-dependent proliferation was observed in control but not in *Piezo1*-deficient MuSCs (Fig S7A–C). These results suggest that PIEZO1 responds to changes in the surrounding mechanical properties, and thus, controls MuSC proliferation.

### *Piezo1* deficiency affects the activation of Rho-GTPase

To reveal the molecular mechanisms underlying the PIEZO1-mediated regulation of MuSCs, we sought to identify the genes whose expression levels were altered by *Piezo1* deficiency. Total RNA was isolated from MuSCs cultured for 3 d in growth medium and subjected to RNA-seq analysis. Interestingly, our RNA-seq analysis detected a large majority of GO terms related to "cytoskeletal components." Furthermore, Gene Set Enrichment Analysis revealed "muscle differentiation" as an up-regulated gene signature in *Piezo1* cKO (Figs 6A–D and S8A). Intracellular signalling for cytoskeletal reorganisation are critical for the function and differentiation of MuSCs, including cell shape changes, migration, and maintenance of stemness (Eliazer et al, 2019). Previous studies have shown that a variety of intracellular signalling cascades are activated in a PIEZO1-dependent manner (Murthy et al, 2017). Thus, we hypothesised that PIEZO1 could act as an upstream activator in MuSCs through Rho-GTPases, which are members of a protein

amplitude of Ca$^{2+}$ fluctuations in MuSCs. Open circles: control; closed circles: *Piezo1* cKO. (>60 cells per condition from N = 3 mice). **(G, H, I)** EdU incorporation assay on cultured MuSCs. **(G)** Time course for chelating cytosolic Ca$^{2+}$ in proliferating MuSCs. **(H)** Representative images of an EdU incorporation assay on MuSCs cultured in media containing DMSO (upper panels) or BAPTA-AM (lower panels). Scale bar: 100 µm. **(I)** Measurement of EdU+ cells; the number of EdU+ cells was normalised to that on control; DMSO (>300 cells per condition from N = 5 mice). Data represent means + SEM. **P < 0.01, ***P < 0.001 (non-paired t test).

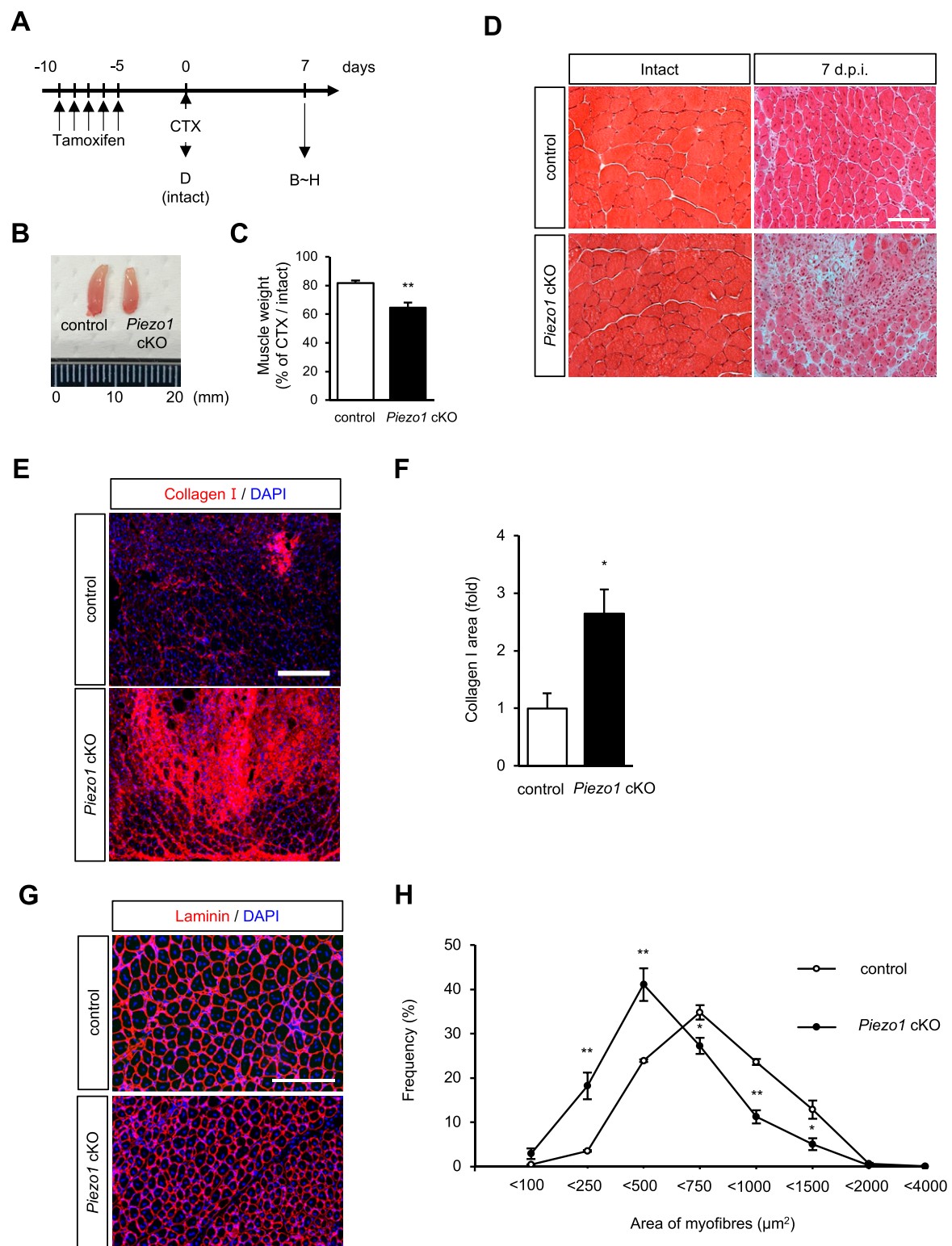

**Figure 3. Impaired regeneration capacity of *Piezo1*-deficient muscle after cardiotoxin (CTX)-induced myofibre degeneration.**
**(A)** Time course for the induction of *Piezo1* deficiency, injection of TA with CTX, and isolation of regenerating muscle samples. **(B)** Representative images of regenerating TA muscle samples in control (left) and *Piezo1* cKO mice. **(C)** Weight of TA muscle samples after CTX-induced muscle injury (N = 3–7 mice per condition). **(D)** Haematoxylin and eosin staining of cross-sections of intact and CTX-injected tibialis anterior muscle samples harvested at 7 d post-injection. Upper panels: control; lower panels: *Piezo1* cKO mice. Scale bar: 100 μm. **(E, F)** Fluorescence intensity of regenerating muscle sections stained with anti-collagen I antibody. The y-axis shows the mean collagen I fluorescence intensity (ratio). Red: collagen I; blue: nuclei (DAPI). Bar graphs represent mean + SEM. White bars: wild-type; black bars: *Piezo1* cKO muscle (N = 3 mice per

family that governs cytoskeletal rearrangement (Vicente-Manzanares et al, 2009). To test this hypothesis, we performed immunofluorescent analysis to detect an active form of Rho and phosphorylated myosin light chain (pMLC), which are essential for the promotion of actomyosin formation. Surprisingly, although the active form of Rho was expected to be up-regulated based on our RNA-seq analysis, immunoreactivity using Rhotekin-RBD (the Rho binding domain in Rhotekin protein), which specifically detects the active form of Rho, revealed reduced amounts of Rho-GTP in *Piezo1* cKO compared with the case in the control (Fig 7A–C). Consistent with this observation, the pMLC expression in MuSCs on isolated myofibres was significantly reduced in *Piezo1* cKO mice compared with that in control mice (Figs 7D and E and S9C–E). The same trend was observed in MuSCs 7 d post-cardiotoxin injection, where impaired regeneration capacity was observed in *Piezo1* cKO mice (Fig S9A and B). Moreover, treatment with CN03 (a Rho activator) rescued the phenotypes observed in *Piezo1* cKO mice, including (i) increased EdU incorporation at the early stages of MuSC activation (Fig 7F and G), (ii) reduction in the MuSC number after MuSC activation on myofibres (Fig 7F and H), and (iii) reduction in the pMLC levels in MuSCs (Fig S9F). Importantly, inhibition of Rho activity using CT04 remarkably reduced the number of Pax7-positive cells on myofibres after MuSC activation (Fig 7I–K). These results collectively indicate that PIEZO1 plays a role in the active Rho-mediated phosphorylation of MLC to regulate MuSC activation and proliferation.

### *Piezo1* deficiency causes mitotic catastrophe in MuSCs

Although *Piezo1* deficiency promoted cell cycle entry at the early phase of MuSC activation, the number of MuSCs in *Piezo1* cKO was less than that in the control in proliferating MuSCs (Figs 4G–J and 5F–H). Therefore, we hypothesised that the M phase of the cell cycle is affected by *Piezo1* deficiency. To test this hypothesis, we examined the localization of PIEZO1-tdTomato during mitosis in MuSCs on isolated myofibres. Surprisingly, PIEZO1-tdTomato was clearly localized in the midbody during cell division of MuSCs (Fig 8A). This result was confirmed by the fact that PIEZO1-tdTomato co-localised with Aurora kinase, Citron kinase, and RhoA (Fig 8B and C), all of which are known to accumulate in the midbody during cytokinesis (Barr & Gruneberg, 2007). This allowed us to further examine the phenotypes associated with *Piezo1* deficiency in MuSCs. Isolated MuSCs were cultured for 2 d in growth medium, until the first cell division took place (Rodgers et al, 2014). Immunofluorescent analysis revealed that although control MuSCs showed successful segregation of chromosomes, *Piezo1*-deficient MuSCs displayed abnormal chromosomal structures such as chromosomal bridges (Fig 8D and E). Moreover, CN03 treatment at least partially rescued the abnormalities during cytokinesis (Fig 8F and G), suggesting that the PIEZO1-Rho pathway may act as a critical determinant for precise cell division, thus enabling MuSCs to proliferate for myogenesis.

## Discussion

Because MuSCs were identified in 1961 (Mauro, 1961), great efforts have been made to understand the molecular mechanisms underlying MuSC-dependent muscle regeneration. Although the mechanical stimulation of MuSCs along with a series of secretory molecules has been thought to play fundamental roles in MuSC functions, the molecular entity that senses changes in the mechanical properties of the surrounding niche remains to be identified. In this study, we report that PIEZO1 is involved in MuSC proliferation by mediating spontaneous $Ca^{2+}$ influx across the plasma membrane. Deletion of *Piezo1* in MuSCs causes a series of phenotypes, including suppressed cell cycle entry, reduced proliferative capacity, and mitotic catastrophe during myogenesis, leading to abnormalities in muscle regeneration.

Spontaneous $Ca^{2+}$ influx is thought to be involved in a variety of physiological signals, especially in non-excitable cells (Qian et al, 2019). Although mechanosensitive ionic currents were first recorded in chicken myoblasts (Guharay & Sachs, 1984), the function of $Ca^{2+}$ influx mediated by mechanosensation during myogenesis is poorly understood. Our results show that $Ca^{2+}$ fluctuations occur in a PIEZO1-dependent manner and that Rho activation is significantly blunted in *Piezo1*-deficient MuSCs, suggesting that $Ca^{2+}$ influx through PIEZO1 may act as an upstream event for Rho-GTPase to regulate the functions of MuSCs. Previous studies have identified several candidate molecules that couple $Ca^{2+}$ influx across the plasma membrane with RhoA-dependent actomyosin reorganisation (Ying et al, 2009; Murakoshi et al, 2011; Pardo-Pastor et al, 2018). Further studies are required to elucidate the molecular mechanisms underlying PIEZO1-mediated Rho activation in MuSCs.

Changes in mechanical properties, such as actomyosin formation, are thought to be involved in MuSC functions. Eliazer et al (2019) demonstrated that myofibre-derived Wnt4 is essential for the maintenance of stemness in MuSCs through the RhoA-dependent repression of the transcription co-activator YAP1. It is possible that PIEZO1 and Wnt4-signalling cooperatively enhance RhoA-mediated actomyosin formation, enabling MuSCs to maintain their functions. Indeed, our study revealed that *Piezo1*-deficiency increased the numbers of Ki67- and EdU-positive MuSCs at the early phase of MuSC activation (Figs 5F and S5B). Moreover, a pharmacological Rho activator completely rescued the phenotype (spontaneous MuSC activation) observed in *Piezo1* cKO (Fig 7D and E), further supporting our hypothesis. Recent studies revealed that the generation of pulses of RhoA activation called "Rho flares" promotes actomyosin-dependent contraction, which is required for maintaining the barrier integrity in epithelial cells (Stephenson et al, 2019; Varadarajan et al, 2022). Interestingly, GsMTx-4, a broad inhibitor of mechanosensitive ion channels including PIEZO1, suppresses the activation of Rho (Varadarajan et al, 2022), suggesting that PIEZO1-Rho signalling could be involved in a variety of biological processes including tight junction remodelling and the promotion of MuSC functions.

---

condition). Scale bar: 100 $\mu m$. **(G, H)** Cross-sectional area of regenerating myofibres at 7 d post-CTX injection. **(G)** Representative images of regenerating muscle sections stained with anti-laminin antibody. **(H)** Measurement of the cross-sectional area in (G) as the percentage of total fibres. Red: laminin I; blue: nuclei (DAPI) (N = 3 mice per condition). Scale bar: 100 $\mu m$. *$P < 0.05$, **$P < 0.01$. (non-paired *t* test). CSA, cross-sectional area.

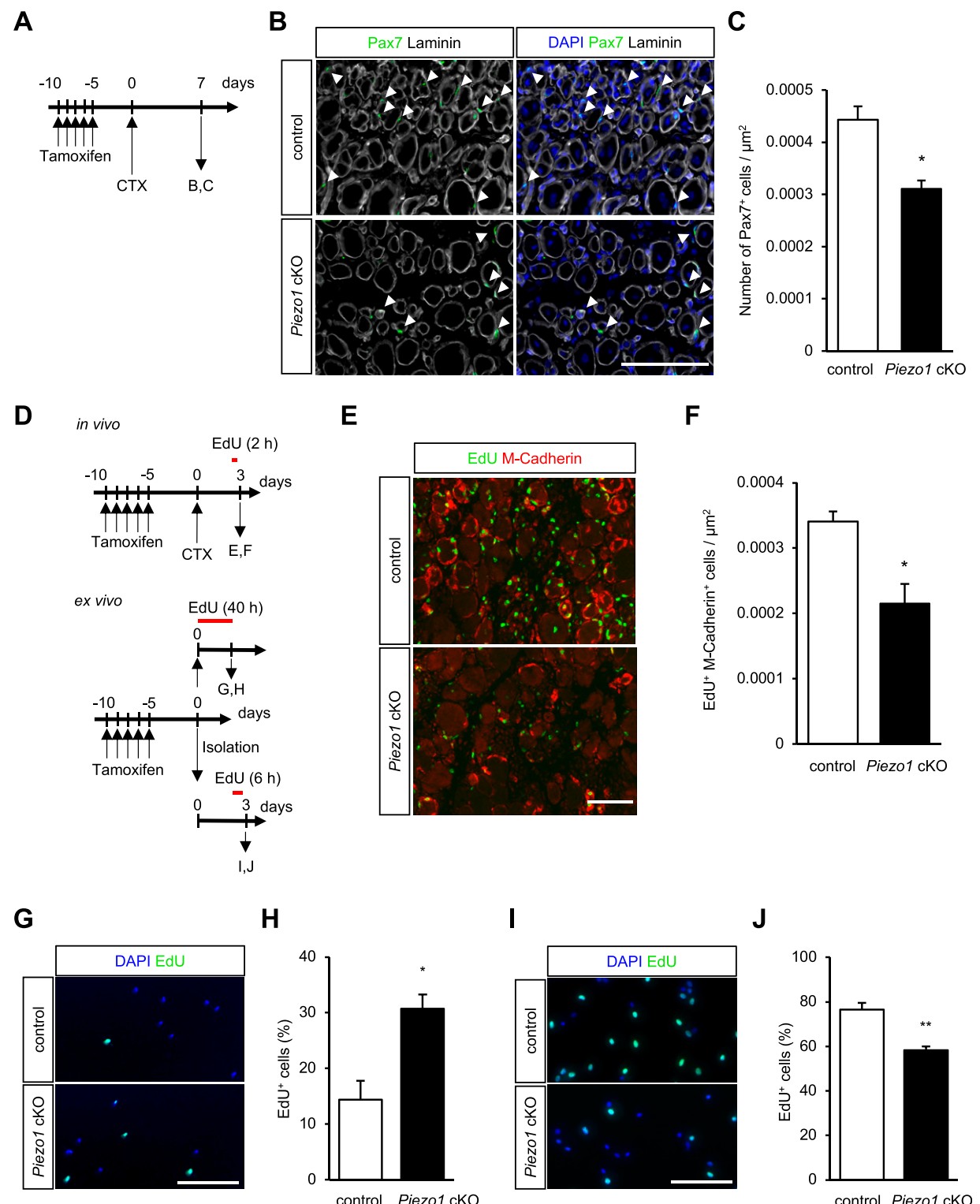

**Figure 4.  Reduced proliferation capacity of muscle satellite cells (MuSCs) with *Piezo1*-deficiency during myofibre regeneration.**
**(A)** Time course for the induction of *Piezo1* deficiency, cardiotoxin (CTX) injection, and harvesting of TA muscle samples. **(B, C)** Detection of Pax7-positive MuSCs (arrowheads) in cross-sections from control (left) and *Piezo1* cKO muscle (right). The number of MuSCs was evaluated 7 d after CTX administration. Scale bar: 100 μm.
**(C)** Measurement of the number of Pax7-positive MuSCs per μm² (N = 3 mice per condition). Bar graphs represent mean + SEM; *P < 0.05. **(D)** Time course of EdU incorporation assays on MuSCs in regenerating TA muscle (in vivo) and isolated MuSCs (ex vivo). **(E, F)** EdU incorporation assays for regenerating the TA muscle (in vivo). After CTX administration, the number of EdU+ M-cadherin+ cells (i.e., MuSCs possessing proliferative capacity) was counted in cross-sections from control and *Piezo1* cKO muscle samples (N = 3 mice per condition). **(E)** Representative images of EdU (green)- and M-cadherin (red)-positive MuSCs in the control (upper panel) and *Piezo1* cKO

Recent studies have shown that PIEZO1 plays a role in MuSC function. Ma et al (2022) demonstrated that MuSCs are morphologically heterogeneous with axon-like protrusions and that PIEZO1 is essential for preserving this feature, thereby priming MuSCs to be more responsive. In this study, we have revealed that *Piezo1* deficiency affects the Rho activation (Fig 7A–C). As Rho signalling acts as a critical determinant for cellular morphogenesis through cytoskeletal reorganisation, the PIEZO1-Rho axis may play a role in the generation of morphological heterogeneity in MuSCs in vivo. Peng et al (2022) showed that PIEZO1 is involved in the p53-mediated ROS signalling pathway. Although our RNA-seq analysis did not show a statistical difference in the up-regulation of p53 signalling pathways (Figs 6 and S7), this could result from the timing of RNA sample preparation. Peng et al used RNA samples from FISC, whereas we analysed cultured MuSCs. Taken together, these results provide insights into PIEZO1-mediated MuSC functions.

We demonstrate that *Piezo1*-deficiency impaired the proliferation of MuSCs. *Piezo1*-deficient MuSCs displayed hallmarks of dysregulated cytokinesis, such as the existence of chromosome bridges (Fig 8E–G), suggesting that PIEZO1 is required for the progression of mitosis. Moreover, PIEZO1 specifically accumulated in the midbody during cytokinesis (Fig 8A–C). It is possible that RhoA could be spatially and temporally activated in a PIEZO1-dependent manner, thereby promoting various aspects of mitosis such as furrow ingression and cytokinesis completion (Lens & Medema, 2019). Further studies on PIEZO1-mediated cell division may improve our understanding of the function of MuSCs during muscle regeneration.

The findings of the present study highlight the stage-dependent functions of PIEZO1 during myogenesis: regulation of MuSC activation and progression of mitosis. Accumulating evidence has shown that under physiological and pathological conditions, the mechanical properties of the MuSC niche are altered (Gilbert et al, 2010; Urciuolo et al, 2013; Cosgrove et al, 2014; Lacraz et al, 2015; Trensz et al, 2015; Quarta et al, 2016; Silver et al, 2021). Despite these findings, the molecular mechanisms by which MuSCs convert biophysical forces into signalling cascades have not been elucidated yet. Along with ion channels, a variety of membrane proteins have been identified as mechanosensors in physiological systems (Vining & Mooney, 2017; Xu et al, 2018; Kefauver et al, 2020). It is tempting to speculate that a series of mechanosensitive proteins have distinct roles in promoting the activation, proliferation, and differentiation of MuSCs. Further studies are needed to unveil the role of mechanosensors in the progression of myogenesis.

# Materials and Methods

## Mice

Animal care, ethical use, and protocols were approved by the Animal Care Use and Review Committee of the Graduate School of Engineering, Kyoto University, and the University of Shizuoka. Sperm samples for transgenic mouse strains, *Piezo1*$^{tm1a(KOMP)Wtsi}$ (called *Piezo1*$^{LacZ}$) and *Piezo1*$^{tm1c(KOMP)Wtsi}$ (West et al, 2015), were purchased from the UC Davis KOMP repository. Cryo-recovery of sperm was carried out by the RIKEN BioResource Research Center (Japan). *Piezo1*$^{tm1c(KOMP)Wtsi}$ mice were mated with *Pax7*$^{CreERT2/+}$ transgenic mice (strain ID:012476; The Jax laboratory; Lepper & Fan, 2010) to generate MuSC-specific *Piezo1*-deficient mice. *Pax7*$^{CreERT2/+}$; *Rosa26*$^{YFP/+}$ mice (Srinivas et al, 2001) were treated with tamoxifen (Sigma-Aldrich) to induce YFP expression in a Pax7-expressing MuSC population. The *Piezo1-tdTomato* mouse line (Ranade et al, 2014) was kindly provided by Dr. Keiko Nonomura.

TMX (dissolved in corn oil at a concentration of 20 mg/ml) was further used to induce Cre recombinase expression. The mice were injected intraperitoneally with TMX at 40 µg/g of body weight for five consecutive days (Figs 2 and 3) and injected with 2 mg TMX daily for 5 d for ex vivo studies.

## MuSC isolation using FACS

MuSCs from uninjured limb muscles were isolated as previously described (Kitajima & Ono, 2018). Briefly, skeletal muscle samples obtained from the forelimbs of mice were subjected to collagenase treatment using 0.2% collagenase type II (Worthington). Mononuclear cells were incubated with allophycocyanin (APC) or PE-conjugated anti-mouse Ly-6A/E (Sca-1) antibody (#122508; BioLegend), APC- or PE-conjugated anti-mouse CD45 antibody (#103106; BioLegend), APC- or PE-conjugated anti-mouse CD31 antibody (#102508; BioLegend), and APC- or PE-conjugated anti-mouse CD106 antibody (#105718; BioLegend) at 4°C for at least 30 min. These cells were resuspended in PBS containing 2% FBS and then subjected to cell sorting to collect CD106-positive cells using MA900 (Sony) or BD FACS Aria II.

## RT-PCR

MuSCs (FISC and GM 4 d) and gastrocnemius muscle were isolated from Pax7-YFP (Kitajima & Ono, 2018) and C57BL/6J mice (10 wk old), respectively. Total RNA was isolated using ISOGEN II (Nippon Gene) or the QIAGEN RNeasy Micro Kit. cDNA was generated using the PrimeScript II first-strand cDNA synthesis kit (Takara). qPCR was performed with PowerUp SYBR Green Master Mix (Thermo Fisher Scientific) using the StepOne system (Thermo Fisher Scientific). Copy numbers were determined using standard curves from *Piezo1* and were compared with 18S ribosomal RNA. Relative expression was calculated using the $2^{-\Delta\Delta Ct}$ method. The primers used are listed in Table S1.

## Ca$^{2+}$ imaging in MuSCs

Fura-2 imaging was performed as previously described with minor modifications (Tsuchiya et al, 2018). For indicator loading, MuSCs were plated on glass-bottomed dishes (Matsunami) coated with Matrigel and incubated with 5 µM Fura-2 AM (Dojindo) at 37°C for

---

(lower panel) sections. Scale bar: 100 µm. **(F)** Number of EdU+ M-cadherin+ cells per µm². **(G, H, I, J)** EdU incorporation assay in cultured MuSCs (ex vivo). EdU was incorporated into MuSCs for 40 h (G, H) before fixation (>300 cells from N = 3 mice per condition) or 6 h (I, J) before fixation (>500 cells from N = 4 mice per condition). Scale bar: 100 µm. *P < 0.05, **P < 0.01 (non-paired *t* test).

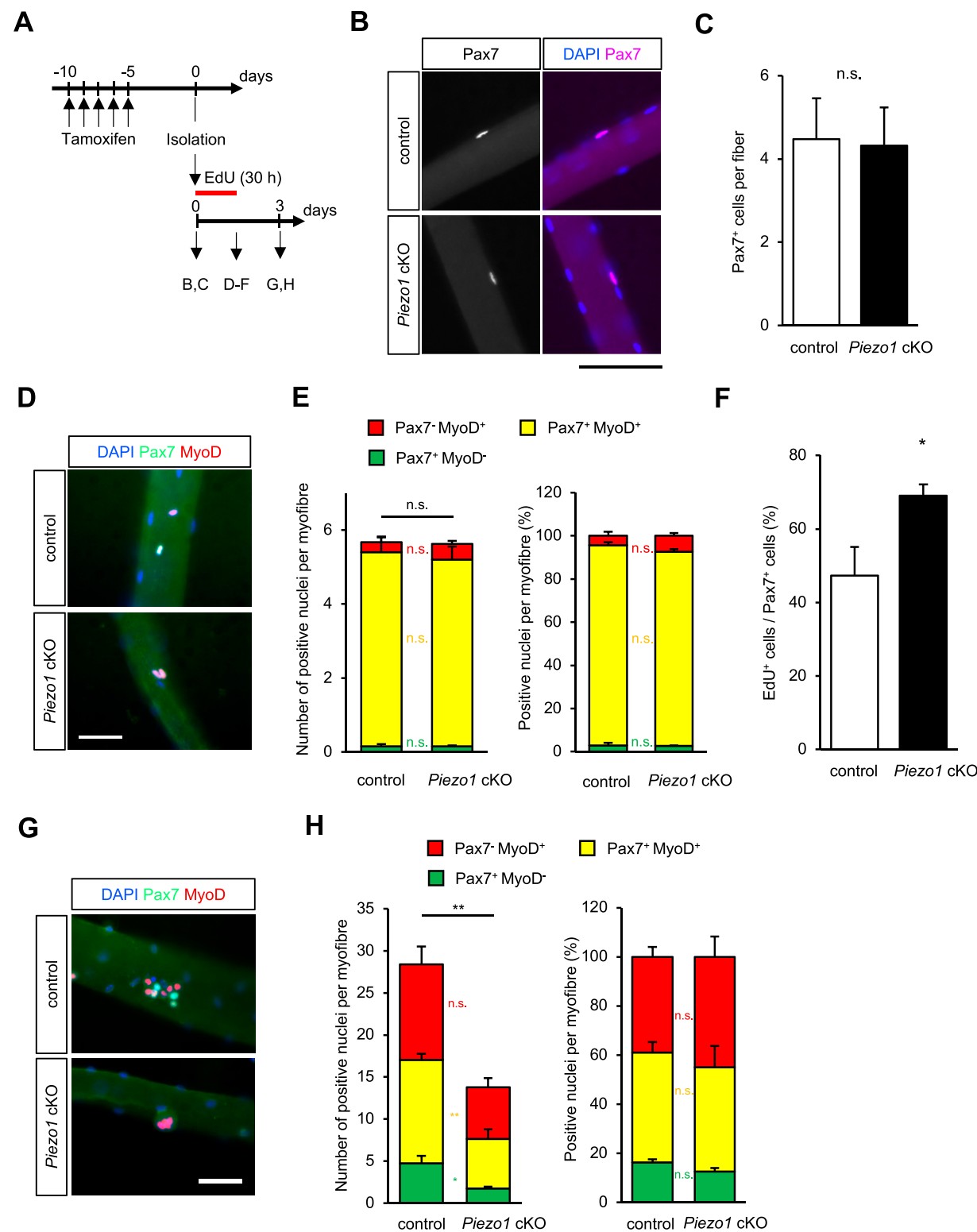

**Figure 5. *Piezo1*-deficiency promotes cell cycle entry of muscle satellite cells (MuSCs) but reduces their proliferative ability on isolated myofibres.**
**(A)** Time course for induction of *Piezo1*-deficiency and harvesting of myofibres. **(B, C)** Detection of Pax7-positive MuSCs on freshly isolated myofibres (open column: control; closed column: *Piezo1* cKO MuSCs) (>15 myofibres per condition from N = 5 mice). Scale bar: 100 μm. **(D, E)** Immunofluorescent analysis of MuSCs on floating myofibres with anti-Pax7 and anti-MyoD antibodies. **(D)** Representative images of Pax7 (green), MyoD (red), and nuclei (blue) on control (upper panel) or *Piezo1* cKO (lower panel) myofibres. Scale bar: 200 μm. **(E)** Measurement of Pax7 and MyoD expression in control and *Piezo1* cKO MuSCs (>15 myofibres per group were investigated from N = 3 mice). **(F)** An EdU incorporation assay on Pax7-positive MuSCs (open column: control; closed column: *Piezo1* cKO MuSCs) (>15 myofibres per condition from N = 4 mice). **(G, H)** Immunofluorescent analysis of MuSCs on floating myofibres with anti-Pax7 and anti-MyoD antibodies. **(G)** Representative images of Pax7 (green), MyoD (red), and nuclei

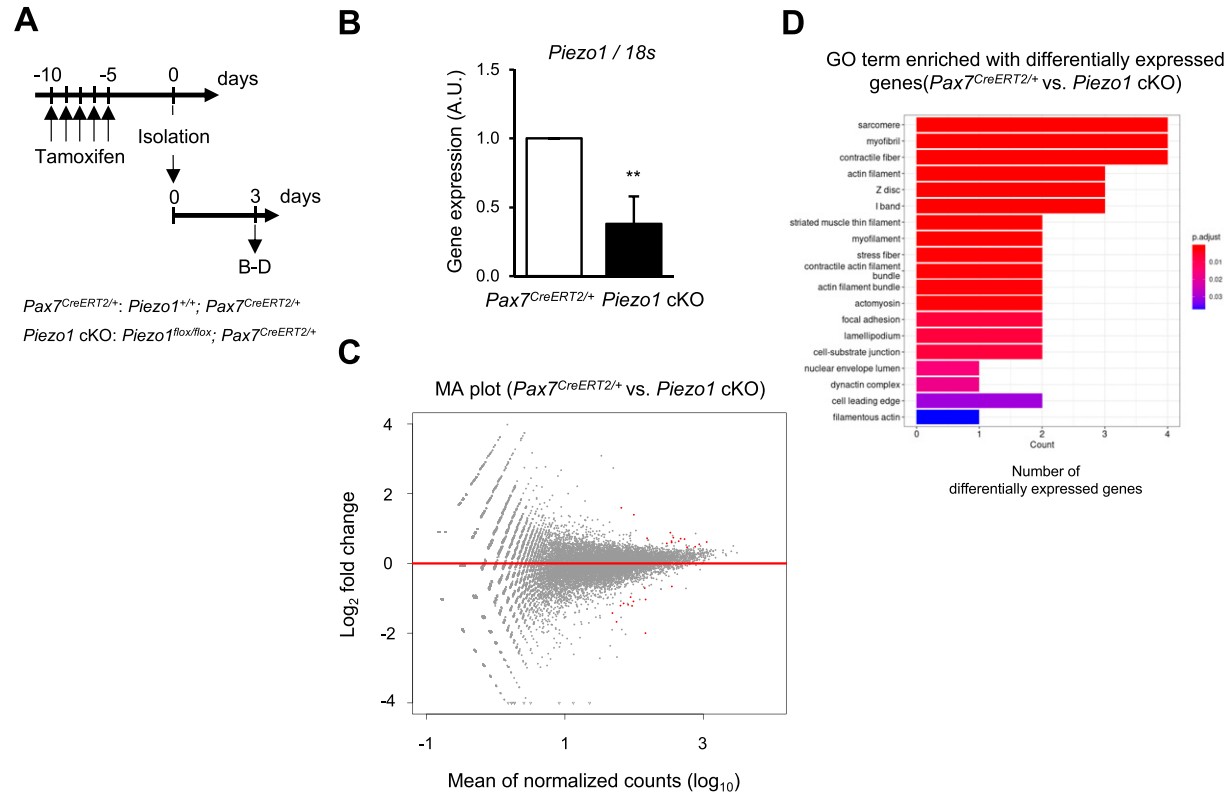

**Figure 6. RNA-seq analysis of *Piezo1*-deficient muscle satellite cells (MuSCs).**
**(A)** Time course for the induction of *Piezo1*-deficiency and isolating MuSCs from control and *Piezo1* cKO muscle (N = 3). **(B)** RT-qPCR analysis of *Piezo1* in MuSCs isolated from control and *Piezo1* cKO. **(C, D)** Transcriptome analysis on total RNA samples of MuSCs isolated from control and *Piezo1* cKO. **(C)** MA plot for visualisation of gene expression difference between control and *Piezo1* cKO. The red dots indicate differentially expressed genes with 15 up-regulated and 12 down-regulated genes. **(D)** Enriched gene ontology terms detected by RNA-seq analysis between control and *Piezo1* cKO. The counts represent the number of up-regulated genes in *Piezo1* deficiency, and the colours of the bars represent the *P*-adjusted values. **P < 0.01 (non-paired *t* test).

60 min. Time-lapse images were obtained every 2 s. The base composition of HBS was (in mM) 107 NaCl, 6 KCl, 1.2 MgCl₂, 11.5 glucose, and 20 HEPES (pH = 7.4 adjusted with NaOH). HBS with 2 mM $Ca^{2+}$ (2Ca) in addition contained 2 mM CaCl₂, whereas that without $Ca^{2+}$ (0Ca) in addition contained 0.5 mM EGTA instead of CaCl₂. Ratiometric images ($F_{340}/F_{380}$) were analysed using Physiology software (Zeiss). Yoda1-induced $Ca^{2+}$ influx was measured as the difference in the Fura-2 ratio between its maximum value and that at 1 min from the initiation of imaging. These experiments were performed using a heat chamber (Zeiss) to maintain the temperature at 37°C throughout the imaging process. The amplitude was calculated using the following formula:

$$\text{Amplitude} = \left(\left[\text{maximum value of the } F_{340}/F_{380} \text{ ratio}\right] - \left[\text{minimum value of the } F_{340}/F_{380} \text{ ratio}\right]\right)/2.$$

### Single myofibre isolation

Myofibres were isolated from the EDL muscles, as previously described (Nagata et al, 2006). Isolated EDL muscle samples were incubated with 0.2% collagenase I (Sigma-Aldrich) in DMEM at 37°C for 2 h. Myofibres were released by gently flushing the muscle samples in the plating medium using a fire-polished glass pipette.

### Cell culture

MuSCs were cultured in growth medium (DMEM supplemented with 30% foetal bovine serum (Sigma-Aldrich), 1% chicken embryo extract (US Biological), 10 ng/ml basic fibroblast growth factor (ORIENTAL YEAST Co., Ltd.), and 1% penicillin–streptomycin (FUJI-FILM Wako Pure Chemical Corporation)) on culture dishes coated with Matrigel (Corning). 4-OH TMX (1 $\mu$M; Sigma-Aldrich) was added to both control and *Piezo1* cKO growth medium for freshly isolated MuSC culture. The reagents BAPTA-AM (20 $\mu$M; Dojindo) and EdU (10 $\mu$M; Life Technologies) were added to the growth medium during culturing. Rho activator II (CN03; Cytoskeleton, Inc.) was added to the growth medium 24 h after plating.

For MuSC growth on floating myofibres, isolated myofibres were cultured in plating medium (DMEM supplemented with 10% horse serum (Gibco), 0.5% chick embryo extract, and 1% penicillin–

(blue) on control (upper panel) or *Piezo1* cKO (lower panel) myofibres. Scale bar: 200 $\mu$m. **(H)** Measurement of Pax7 and MyoD expression in control and *Piezo1* cKO MuSCs (>15 myofibres per condition from N = 5 mice). *P < 0.05, **P < 0.01, n.s., not significant. (non-paired *t* test).

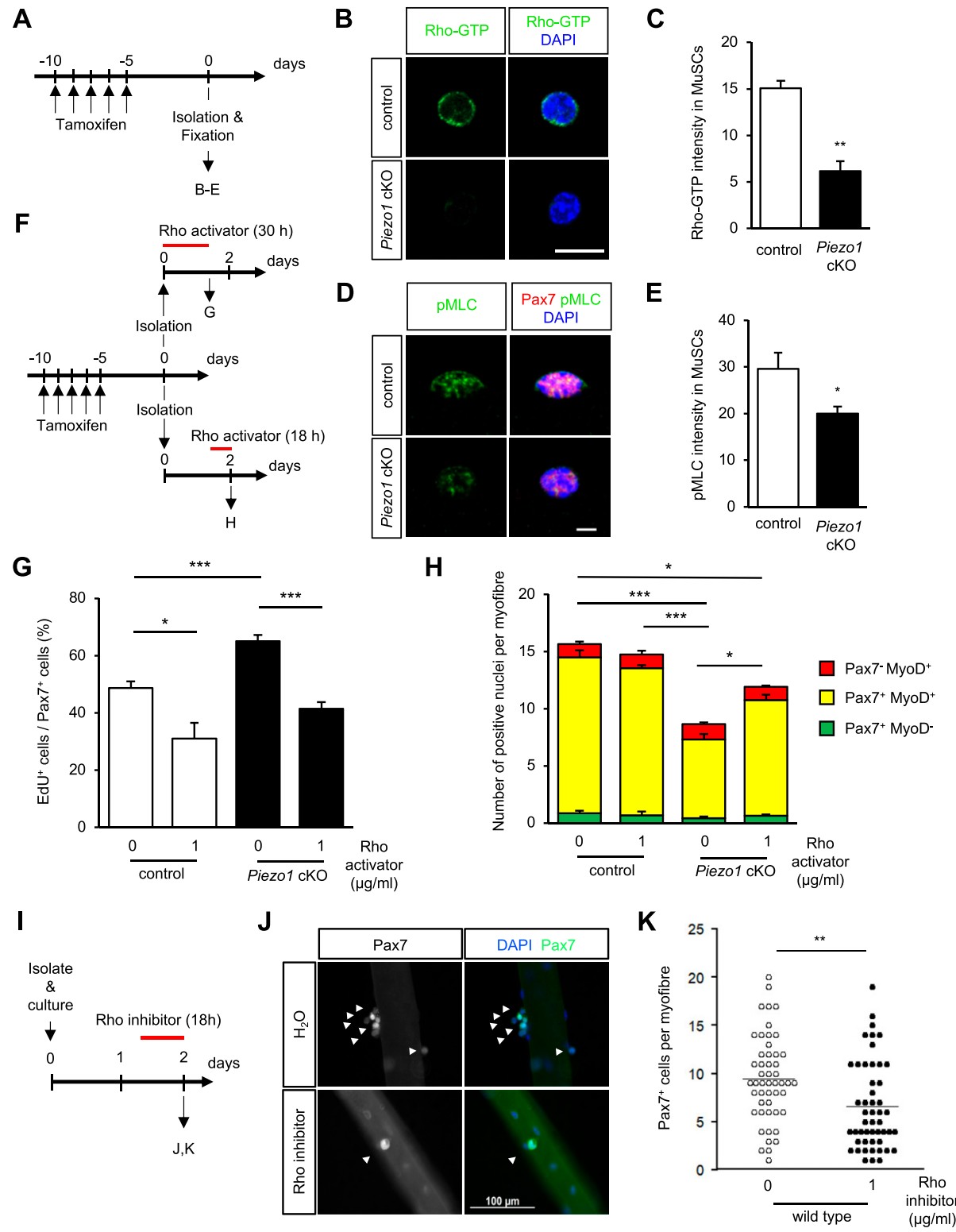

**Figure 7. Reduced activation of the Rho-pMLC signalling in *Piezo1*-deficient muscle satellite cells (MuSCs).**
**(A)** Time course for induction of *Piezo1*-deficiency and isolation of MuSCs. **(B, C)** Detection of an active form of Rho-GTP in freshly isolated MuSCs. **(B)** Immunofluorescent analysis for detection of Rho-GTP (Green) using Rhotekin-Rho binding domain. **(C)** Measurement of the fluorescence intensity of Rho-GTP (>80 cells per condition from N = 3–4 mice). Bar represents means + SEM. Scale bar: 10 μm. **(D, E)** Detection of phosphorylated MLC (pMLC) in MuSCs on myofibres from intact muscle. **(D)** Immunofluorescent analysis of MuSCs. Pax7 and pMLC were detected with anti-Pax7 and anti-pMLC antibodies, respectively. **(E)** Measurement of the fluorescence intensity of pMLC (>30 cells per condition from N = 5). Bar represents means + SEM. Scale bar: 10 μm. **(F, G, H)** Effects of a Rho activator CN03 on the activation and proliferation of control or *Piezo1* cKO

streptomycin) at 37°C with 5% $CO_2$. Next, 5 $\mu$M EdU (Life Technologies) was added to the plating medium to analyse S-phase entry. Rho activator II (CN03; Cytoskeleton, Inc.) or Rho inhibitor I (CT04; Cytoskeleton, Inc.) was added to the plating medium at indicated time points shown in Figs 7F and I and S9C.

## MuSC culture on matrices with different elasticities

Freshly isolated MuSCs were plated on CytoSoft Imaging 24-well Plates (2 kPa: 5185-1EA; 32 kPa: 5188-1EA; Advanced BioMatrix) coated with PureCol Type I Collagen (No. 5005-100ML; Advanced BioMatrix), based on the manufacturer's instructions.

## Immunofluorescent analysis

Cells were added to Matrigel-coated glass-bottom dishes and fixed with 4% PFA/PBS for 10 min. After permeabilisation in 0.1% Triton X-100/PBS for 10 min, samples were blocked in 1% BSA/PBS for 1 h and probed with primary antibodies (Table S2) at 4°C overnight. After multiple washes with PBS, the suitable secondary antibodies (Table S2) were added to the samples. Nuclei were detected using DAPI (1:1,000; Dojindo). For RhoA detection during cytokinesis, MuSCs were fixed with 10% trichloroacetic acid on ice for 15 min. They were then rinsed twice with PBS containing 30 mM glycine (G-PBS) and treated with 0.2% Triton X-100 in G-PBS for 5 min for permeabilisation. After blocking with 1% BSA/PBS, cells were incubated in primary antibodies (Table S2) at 4°C overnight.

Myofibres were fixed with 2% PFA/PBS for 5 min. After permeabilisation and blocking with 0.1% Triton X-100 in 1% BSA/PBS for 15 min, the samples were probed with the antibodies (Table S2) at room temperature for 2 h or at 4°C overnight. After washing once with PBS, suitable secondary antibodies (Table S2) were added. Nuclei were detected using DAPI (1:1,000; Dojindo). Immunofluorescent signals were visualised with Alexa 488- or Alexa 555-conjugated secondary antibodies using an epifluorescence microscope (Axio Observer; Z1; Zeiss) with 10×, 20×, 40× objective lenses or a confocal microscope (LSM 800; Zeiss) with a 63× objective lens. The fluorescence intensity was measured using ImageJ software for statistical analyses.

For EdU detection, a click chemical reaction was performed after primary and secondary staining using a Click-iT EdU Imaging Kit (Life Technologies) according to the manufacturer's instructions.

## In situ binding assay for GTPase activity

Freshly isolated MuSCs were plated on glass-bottomed dishes (Matsunami) coated with Matrigel and immediately fixed with 4% PFA for 25 min. The MuSCs were then washed with PBS twice and permeabilised with 0.5% Triton X100/PBS for 10 min. They were blocked in 5% FBS/PBS and incubated for 1 h at room temperature with GST tagged Rhotekin-Rho binding domain (Cytoskeleton Inc.). After washing with PBS twice, cells were incubated in anti-GST Alexa Fluor 488 (1:500 dilution; Invitrogen) and DAPI in 5% FBS/PBS for 1 h at room temperature, before imaging.

## Histological analysis

Cardiotoxin experiments were performed as previously described (Tsuchiya et al, 2018). Cardiotoxin (50 $\mu$l of 10 $\mu$M; Latoxan) was injected into the TA muscle of 8- to 15-wk-old mice. The muscle was harvested at the time points indicated in each figure and snap-frozen in isopentane cooled with liquid nitrogen. Cross-cryosections (thickness, 7 $\mu$m) of the muscle samples were used for haematoxylin and eosin staining, as previously described (Hara et al, 2011). CSA in each muscle sample was determined using cryosections stained with anti-laminin I antibody (L9393; Sigma-Aldrich; 1:500). The fibrotic area was detected using an anti-collagen I antibody (1310-01; Southern Biotech; 1:500). The CSA and fluorescence intensity were measured using the ImageJ software for statistical analyses.

## In vivo EdU-uptake assay

EdU was dissolved in PBS at 0.5 mg/ml and injected intraperitoneally at 0.1 mg per 20 g body weight at the time points indicated in each figure.

## RNA-seq analysis

For RNA-seq analysis, total RNA was obtained from cultured MuSCs using QIAGEN RNeasy Micro Kit. Quality and concentration of the total RNA were assessed using Bioanalyser (Agilent RNA 6000 Pico Kit; Agilent Technologies). To synthesize cDNA, 500 pg of the total RNA were reverse transcribed with SMARTScribe Reverse Transcriptase (639536; Takara Bio) in a 15 $\mu$l reaction volume, following the manufacturer's protocol with minor modifications. Briefly, the reaction mix contained 1× First-Strand Buffer, 2.4 $\mu$M biotinylated LNA template-switching oligonucleotide (TSO: AAGCAGTGGTATCAACGCAGAGTACrGrG+G; QIAGEN), 1.2 $\mu$M barcoded primers (Table S3), 1 mM dNTP mix, 2 mM DTT, 11 U/$\mu$l RNase inhibitor, 1 U/$\mu$l SMARTScribe Reverse Transcriptase (639536; Takara Bio), and 0.475× Lysis buffer (365013; Takara Bio). The reaction mix was incubated at 42°C for 90 min and then 70°C for 10 min. The residual primers were digested by adding 0.75 $\mu$l of exonuclease I (5 U/$\mu$l) and incubating at 37°C for 30 min followed by 80°C for 20 min. The synthesized cDNA was amplified by 12 cycles of PCR with SeqAmp DNA polymerase (638504; Takara Bio). The PCR product was purified with 1.0×

MuSCs on myofibres. **(F)** Time course for the induction of *Piezo1*-deficiency, administration of Rho activator, and myofibre harvesting. **(G)** An EdU incorporation assay on Pax7-positive MuSCs (open column: control; closed column: *Piezo1* cKO MuSCs) (>15 myofibres per condition from N = 4 mice). **(H)** Evaluation of Pax7 and MyoD expression in control and *Piezo1* cKO MuSCs (>15 myofibres per condition from N = 4 mice). **(I, J, K)** Effect of the Rho inhibitor CT04 on proliferation of control or *Piezo1* cKO MuSCs on myofibres. **(I)** Time course of Rho inhibition during MuSC proliferation on myofibre. **(J)** Immunofluorescent analysis of MuSCs on floating myofibres with anti-Pax7 after Rho inhibition. **(K)** Effect of a Rho inhibitor on the proliferation of wild-type MuSCs (>50 myofibres from N = 3 mice). *$P < 0.05$, **$P < 0.01$, ***$P < 0.001$. ((C, E, K) non-paired $t$ test, (G, H) Tukey's test).

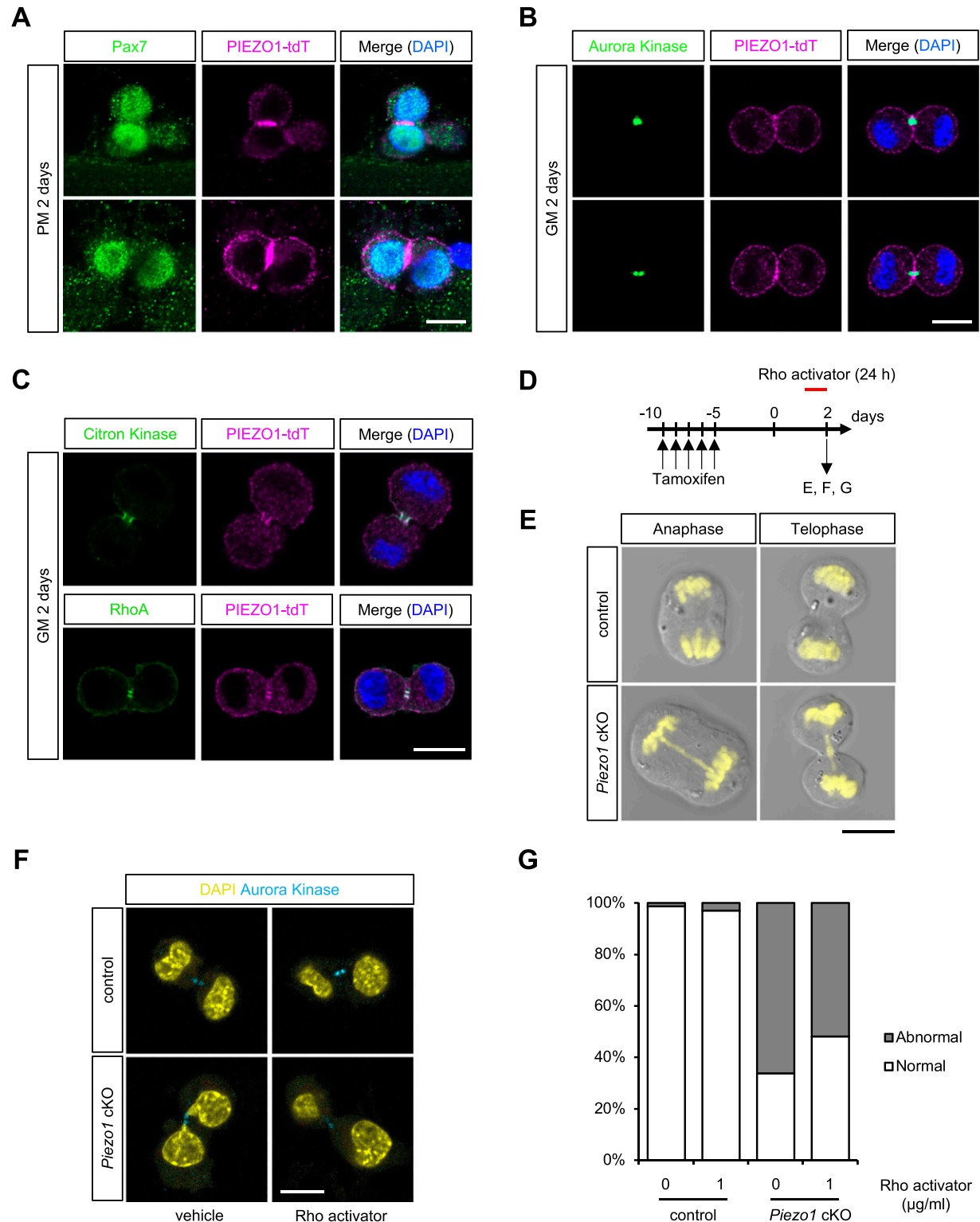

**Figure 8. Mitotic catastrophe in muscle satellite cells (MuSCs) isolated from *Piezo1* cKO.**
**(A)** Detection of PIEZO1-tdTomato in mitotic MuSCs on myofibres isolated from *Piezo1-tdTomato*. Pax7 and nuclei were co-stained with anti-Pax7 antibody and DAPI, respectively. Scale bar: 10 $\mu$m. **(B, C)** Co-localisation of PIEZO1-tdTomato with cytokinetic components in MuSCs isolated from *Piezo1-tdTomato*. Aurora kinase (B), Citron kinase and RhoA (C), were detected with specific antibodies. Scale bar: 10 $\mu$m. **(D, E, F, G)** Mitotic catastrophe in *Piezo1* cKO MuSCs. **(D)** Time course for the induction of *Piezo1*-deficiency and administration of the Rho activator. **(E)** Evaluation of chromosomal integrity during cell division in control and *Piezo1* cKO. Chromosomal DNA was detected with DAPI. **(F, G)** Rescue of chromosomal catastrophe observed in *Piezo1* cKO MuSCs with a Rho activator CN03 (>50 cells per condition from N = 3–4 mice were investigated). Scale bar: 10 $\mu$m.

AMPure XP (Beckman Coulter). The purified DNA samples were pooled at an equal mass concentration and 400 pg was used for constructing a sequencing library (Nextera XT DNA Library Preparation Kit; Illumina). The final product was purified with 0.6× of AMPure XP. The sequence library was analysed on an Illumina HiSeqX platform with paired-end reads of 150 bp. Mapping of sequence reads to a reference genome (GRCm38.102) was done using STAR (2.7.9a). The mapped reads were annotated by featureCounts (v.2.0.1) and counted by UMI-tools (ver.1.1.1). Count data were analysed using R (version 4.0.0) and the DESeq2 version 1.30.1 using the default parameters and then MA-plot was also obtained. For each pairwise comparison, raw *P*-values were adjusted for multiple testing according to the Benjamini and Hochberg (BH) procedure (Benjamini & Hochberg, 1995) and genes with an adjusted *P*-value lower than 0.05 were considered differentially expressed. Gene ontology analyses and gene set enrichment analyses were performed using clusterProfiler with the cellular component option.

### Western blot analysis

For Western blot analysis, the proteins were separated by SDS-PAGE and transferred onto a polyvinylidene difluoride membrane. The membrane was blocked in TBS containing 1× EzBlockChemi (ATTO) and 0.1% Tween 20, stained with antibodies listed in Table S2, and visualised using an enhanced chemiluminescence reagent (SuperSignal West Pico PLUS substrate; Thermo Fisher Scientific).

### Statistical analysis

Statistical analyses were performed using Microsoft Excel or JMP 11 (JMP Statistical Discovery LLC). The statistical significance of the differences between the mean values was analysed using a non-paired *t* test (two-sided). Multiple comparisons were performed using Tukey's test followed by ANOVA. *P* values of \*$P < 0.05$, \*\*$P < 0.01$, and \*\*\*$P < 0.001$ were considered statistically significant. The results are presented as the mean + SEM. n.s. indicates results that are not statistically significant.

## Data Availability

All data generated and analysed in this study are available from the corresponding author on reasonable request. The custom compute codes used to generate results in this study are available from the corresponding author on reasonable request. Any additional information required to reanalyze the data reported in this paper is available from the corresponding author upon request. The RNA-seq data from this publication has been deposited to the GEO database (https://www.ncbi.nlm.nih.gov/geo/; accession number: GSE217417).

## Supplementary Information

## Acknowledgements

We thank Drs. Hiroshi Takeshima and Atsuhiko Ichimura (Kyoto University) for helpful discussion; Dr. Itaru Hamachi (Kyoto University) for technical support; and members of the Hara laboratory for their scientific contributions. This study was supported by the Japan Agency for Medical Research and Development (AMED, JP19gm5810016); the Grant-in-Aid for Scientific Research KAKENHI (19H03179, 22H03484); Intramural Research Grant (2-5) for Neurological and Psychiatric Disorder of NCNP; grants from Takeda Science Foundation, Ohsumi Frontier Science Foundation, Nakatomi Foundation, and the Asahi Glass Foundation (to Y Hara); and Grant-in-Aid for JSPS Research Fellow (20J22984 to K Hirano)

## Author Contributions

K Hirano: conceptualization; data curation; formal analysis; investigation; methodology; and writing—original draft, review, and editing.
M Tsuchiya, E Umemoto, and Y Ono: resources and methodology.
A Shiomi: software and methodology.
S Takabayashi and K Nagao: methodology.
M Suzuki, Y Ishikawa, Y Kawano, and Y Takabayashi: investigation.
K Nishikawa: data curation and investigation.
Y Kitajima: resources, investigation, and methodology.
K Nonomura: resources.
H Shintaku: resources, software, investigation, methodology, and writing—review and editing.
Y Mori and M Umeda: conceptualization and supervision.
Y Hara: conceptualization; supervision; funding acquisition; project administration; and writing—original draft, review, and editing.

## Conflict of Interest Statement

The authors declare that they have no conflict of interest.

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
