## [Reviewer comments · Life Science Alliance]

Life Science Alliance

The mechanosensitive ion channel PIEZO1 promotes satellite cell function in muscle regeneration

Kotaro Hirano, Masaki Tsuchiya, Akifumi Shiomi, Seiji Takabayashi, Miki Suzuki, Yudai Ishikawa, Yuya Kawano, Yutaka Takabayashi, Kaori Nishikawa, Kohjiro Nagao, Eiji Umemoto, Yasuo Kitajima, Yusuke Ono, Keiko Nonomura, Hirofumi Shintaku, Yasuo Mori, Masato Umeda, and Yuji Hara

DOI: <https://doi.org/10.26508/lsa.202201783>

Corresponding author(s): Yuji Hara, University of Shizuoka

Review Timeline:

Submission Date:	2022-10-23
Editorial Decision:	2022-10-24
Revision Received:	2022-10-28
Editorial Decision:	2022-11-03
Revision Received:	2022-11-08
Accepted:	2022-11-08

Transaction Report:

Please note that the manuscript was previously reviewed at another journal and the reports were taken into account in the decision-making process at Life Science Alliance.

Referee #1 Review

Report for Author:

In this paper titled "The mechanosensitive Ca²⁺-permeable ion channel PIEZO1 promotes satellite cell function in skeletal muscle regeneration", the authors showed that PIEZO1, a bona fide mechanosensitive ion channel, promotes the proliferative and regenerative function during skeletal muscle regeneration using conditional knockout mice. Although first time characterizing the MuSC function with a genetic mouse model, the study suffers from some fundamental issues in experimental design and data interpretation, I also don't find the findings of the phenotypes or the underlying mechanism carry enough novelty to warrant the publication in this journal.

Major issues:

1. Fig. 1A, the QSC should be called Freshly isolated SCs (FISCs) as it is an acceptable notion in the field of MuSC that FASC sorted cells are not real quiescent cells. The whole muscle tissue is used as a substitute for myofiber, which is not appropriate. The authors should examine Piezo1 expression profiles in the entire lineage development eg. QSC, FISC, ASC-24, ASC-48, ASC-72...DSC...confirm it is indeed highly induced in proliferating myoblasts but decreased in differentiating myofibers. This can also be done by analyzing the published RNA-seq data from various papers. In addition, in Fig 1A, it is known that the expression of Gapdh is induced during SC activation, so alternative controls such as 18S should be used to examine Piezo1 expression by RT-PCR. Moreover, quantitation of the semi-quantitative RT-PCR from at least 3 replicates should be performed.
2. Fig. 1B, Western blotting should be performed in samples from the above mentioned time points of SC lineage progression to confirm its expression dynamics. In addition, there is no description of Fig 1C or EV1B in the Result.
3. I have major issues with Fig. 2, have the authors measured the deletion efficiency of Piezo1 in SCs from the cKO mice? It should be evaluated on DNA, mRNA and protein levels. When were the SCs isolated after tamoxifen injection? In Fig 2A, what does each line of different color stand for? Also, the legend writing is not precise, 2Ca is not in the "upper panel" but "left panel". In EV2A, what do Piezo1Tm1a/Tm1C/Tm1d indicate? Why were 2 different chelators (EGTA and BAPTA-AM) were used? What is the difference between these two? In Figure 2G, representative images of the EdU staining should be shown.
4. In Fig 2C, it is known that the use of Pax7CreERT2/+ causes heterologous deletion of the Pax7 gene, so Piezo1flox/flox; Pax7+/- is not an appropriate control for the cKO mice; Instead, Pax7CreERT2/+ littermates should be used as controls.
5. The characterization of the mutant phenotype in Fig. 3 should be more completed for example by sampling muscles from different time points after CTX injection. The evaluation of fiber size should be done by showing the range of fiber size distribution instead of the average fiber size. Staining of eMyHC should also be performed to evaluate the degree of muscle regeneration. Also in Fig EV4B, it looks like the fibers in Pax7CreERT2/+ are evidently smaller in size compared to the control, but the statistic result shows no difference.
6. In Fig 6, Western blot should be performed to detect the change of pMLC after Piezo1 deletion. In Fig 6D, the authors should explain why CN03 treatment caused change of SC number in wide type group which was opposite to that of Piezo1 deletion group. Moreover, the author should prove whether CN03 treatment would rescue the abnormalities caused by Piezo1 deletion. Overall, I don't think the presented mechanism carries sufficient novelty considering very similar pathway has been shown in their previous paper using C2C12 myoblasts.
7. The writing also needs to be polished. In Discussion, instead of interpreting their data, the authors write too much on mechanics sensing of MuSC and niche environment, which was not even studied in the Result. In a couple of places, I don't think the phrases used are precise. For example, on page 4 bottom, "However, the function of PIEZO1 in MuSC remains to be

elucidated", this is not a precise justification for their study. On page 10, the use of "cell fate decision" is not precise in interpreting their related result.

Referee #2 Review

Report for Author:

The mechanosensitive Ca²⁺-permeable ion channel PIEZO1 promotes satellite cell function in skeletal muscle regeneration. In this report, Hirano and colleagues are investigating the role of the mechanosensory calcium channel Piezo1 in muscle stem cell regeneration. This study addresses a very pertinent question, which is the way by which quiescent stem cells sense and transduce mechanical stimuli from the microenvironment when homeostasis is perturbed. Using a mouse model in which Piezo1 is specifically and conditionally deleted in MuSCs, the authors show that this gene is required for their proliferation and efficient muscle repair. Mechanistically, Piezo1 is suggested to mediate the phosphorylation of Myosin Light chain through the Rho/Rack pathway, which can act on the rearrangement of the cytoskeleton. Overall, this study is interrogating an interesting biological process, it is well conducted and is using an appropriate mouse model to investigate the role of Piezo1 in vivo. However, some aspects of this work remain preliminary and have to be more rigorously addressed for publication.

Specifically:

1. Based on published transcriptomic data, Piezo2 is also expressed in MuSCs. It has been shown that Piezo proteins can synergistically act in transducing mechanical signals and are involved in compensatory mechanisms. It would be interesting to complement Fig.1A by showing the impact of Piezo1 deletion on Piezo2 transcripts.
2. Referring to Fig.1A, it is written "Piezo1 mRNA expression was clearly and moderately detected in QSCs and ASCs"; if anything, Fig.1A shows more Piezo1 in ASC. In any case, RT-qPCR, instead of semi-quantitative RT-PCR, should be performed to draw accurate quantitative conclusions.
2. A recent study (Eliazer et al. CSC 2019) demonstrated that a Wnt4-Rho-Rock axis is required for MuSC quiescence maintenance. This study also showed that genetic inhibition of Rho induced expansion of the MuSC pool (increase of Pax7+/BrdU+ cells) and MuSC activation (increase of Myod+). Given the link between Piezo1 and Rho, and the fact that Piezo1 is expressed in freshly isolated MuSCs (Fig. 1A), it would be interesting to investigate if Piezo1 cKO have an effect on the quiescent MuSC in resting muscle. To address this, the authors could score Pax7 cells (as well as for ectopic proliferation (EdU) and Myogenin cells) a few weeks (usually 3 to 4 weeks) after tamoxifen inductions.
3. If no phenotype is found in quiescent Piezo1 cKO MuSCs, given the sensor property of Piezo1, it would be of great interest to challenge the cKO mice on a treadmill and see how the mutant MuSC respond to mechanical stimuli.
4. Changes in cytoskeletal signaling and mechanical properties are early events during the transition from quiescence to activation. Consequently, it is difficult to estimate if the changes in pMLC are a consequence of activation or a direct consequence (direct target?) of Piezo1 deletion. Thus, the manuscript would be significantly improved by elucidating on the mechanism downstream of Piezo.
 - The authors should perform Western blot analysis of the levels of pMLC vs MLC in Control and Piezo cKO MuSCs, to show that the decrease of pMLC is not due to a decrease of total MLC protein.
 - Does Rho activation rescue the phosphorylation of MLC in Piezo cKO MuSCs?
 - What is the status of Rho mRNA and/or protein upon Piezo cKO? Is Rho less expressed or less functional (to phosphorylate pMLC)?
 - Does Rho inhibition (EV.5E) impact pMLC?
6. Fig.3B: Based on the images of this Figure, there seems to be a "homing" phenotype in the mutant, where the Pax7 cell are blocked in the interstitial space and not located under the basal lamina. It is not easy to estimate that based on a single picture, but I strongly recommend that the authors quantify this potentially interesting phenotype.
7. Fig.3D-E. The quantification does not correlate well with the pictures (or vice versa). It is also unclear how the authors define the "Area of regenerating fibers (graph axis label)".
8. Please, provide statistics between datasets 1-3 (is the no. of Pax7 different in the cKO?) and 2-4 (is the Rho activator restoring the Pax7 cells?)
9. Fig.4D-F: The lower EdU observed in the mutant MuSC, is it due to a delay at the cell cycle entry? Given the proposed role of Piezo1 as a sensor of environmental stress, it would be relevant to assess if this gene has an impact on the activation time and entry to the cell cycle. This could be assessed by performing EdU chase experiments on freshly isolated MuSCs for 40-45h to capture the first S-phase.
10. The authors have analyzed muscle sections on D7 post-Ctx injury and have concluded that "conditional deletion of Piezo1 in MuSCs delays myofiber regeneration after myofiber injury". It is strongly recommended that they also analyze the regeneration

at a later time point, like D28 post-Ctx, in order to determine whether the repair defect upon Piezo1 ablation is maintained or restored by potential compensatory mechanisms.

11. Fig. 1C. Quantification is needed. Do all MuSC express PIEZO1-dtTom? In fact, on page 6, 2nd paragraph the authors refer to detection of Piezo1-dtTom based cytometry, however, the data are not presented. It is recommended to include the FACS analysis with the corresponding quantifications in a figure.

12. In Figure 2C the conditional Piezo1 mice are introduced. Technically, this is not the appropriate control and it would be better to use Pax7-CT2; Piezo+/+ treated with tamoxifen. Also, you should show efficiency of Piezo1 deletion by RT-qPCR on FACS-isolated MuSCs.

13. EV.1C: Piezo1-dtTom seems to also be expressed in Pax7-neg cells. What is the expression pattern of Piezo1 during myogenic commitment (in Myogenin+/Pax7- cells and freshly formed myotubes)?

Editorial comments:

i. The authors refer to freshly isolated MuSCs as quiescent, however, recent studies have shown that these cells are activated during the dissociation process. Please, change the designation from QSC to freshly-isolated or early-activated MuSCs.

ii. Fig.1B, change labels from PIEZO1 to PIEZO1-dtTom. In the current format it is misleading.

iii. Rephrase the title. Satellite cell function is too broad and vague description. It could be changed to "proliferative and regenerative function"?

iv. At the first sentence of the Introduction it is written "muscle regeneration after injuries induced by repeated contraction and relaxation of myofibers". Is this true? Injuries in general are linked to myofiber necrosis. Can the authors include references describing contraction/relaxation-induced regeneration and MuSC activation?

v. In Figure 2C the conditional Piezo1 mice are introduced. Please, consider the following recommendations: The mice indicated as "wild-type" are in fact Piezo flox/flox. Please, change to "Control".

vi. EV.1B is not cited in the main text.

Referee #3 Review

Report for Author:

In the current manuscript Hirano et al describe a function of the mechanosensitive calcium channel Piezo1 in skeletal muscle satellite cell (MuSC) function. They demonstrate expression Piezo1 in MuSCs, reveal that Piezo 1 supports spontaneous calcium entry into MuSCs and show that deletion of Piezo1 impairs proliferation of MuSCs. In addition they demonstrate that deletion of Piezo1 in MuSCs impairs myofiber regeneration after injury. The findings are interesting however interpretation of the results still is limited by some shortcomings. The paper focuses on the description of the MuSC-specific Piezo1 knock-out and might demonstrate the importance of Piezo1 for MuSC function, but the exact mechanisms of Piezo1 action in MuSCs remain largely unexplored.

Specific comments:

Quantification of peaks observed in WT MuSCs with and without calcium (Fig. 2B) seems not to match the representative traces depicted in Fig. 2A. The black trace in Fig. 2A has a peak amplitude of > 0.1 , however there is no such point in the quantification in Fig. 2B. The description of how the quantification is performed is poor. There should be a clear description of how this quantification was performed (e.g. highest peak per xx sec trace). In the figure legend, there should be a description of how many individual cell isolated from how many animals were observed, and it should be made clear how peaks were identified and how the amplitude of the peaks was determined. Based on this, the statistical test used for the quantified data needs to be carefully considered. This criticism similarly applies to Fig. 2D-E. The statement in the figure legend "data represent means +/- SEM" is wrong, in the figure 2 B/E there are no errorbars but datapoints plus a bar indicating means or median. There corresponding author should consult the experimenters for a better description of the data.

EdU positive MuSCs were counted after application of BAPTA-AM in vitro. Representative pictures of those staining should be included into the manuscript and the absolute numbers or percentage of EdU incorporating cells should be included to allow judgment of the proliferative activity of those cells in culture. The authors need to indicate the number of animals used for isolating MuSCs, how many cells were counted per animal, or how many independent experiments were performed in the figure legend to reveal the statistical basis of the bar graphs.

To confirm the CTX injury the authors should add pictures to figure 3 that clearly demonstrate centrally localized nuclei in the regenerating muscle tissue. The statement on infiltrating immune cells in the results section needs to be qualified by appropriate IF staining or by any other means. Again, the statistical basis of the bar graphs need to be clarified. The authors should carefully

discuss all possible explanations for the reduced regeneration of myofibers after CTX injection.

The number of Pax7 positive MuSCs was quantified in muscle sections and seems to be reduced in Piezo1 KO mice. The authors also should use and quantify additional markers of quiescent and proliferating MuSCs in combination with Pax7 staining to demonstrate the physiological state of the MuSCs in vivo. In addition, the authors need to include Pax7-CreERT pos controls to Fig. 4A-C. The number of positive cells should be indicated per μm^2 . The type of muscle used for Fig. 4A-C should be indicated in the figure legend and at least one different muscle should be included in this analysis for a more general statement on the MuSC population of the skeletal muscle.

EdU incorporation into MuSCs was quantified after CTX injury; representative pictures of sections used for quantification should be included into the figure. There is massive, non-physiological stimulation of proliferation after CTX injury. To explore why there are less MuSCs in resting muscle, data on EdU incorporation in undisturbed muscle should be included into the analysis.

The number of Pax7 positive cells on isolated myofibers is not reduced. On the other hand, there is a reduced number of MuSCs / view field in muscle sections. This is an obvious contradiction and the authors need to explain this. One possible explanation would be a different size/number of myofibers. The number of Pax7 positive cells per fiber appears quite high; the authors should include representative pictures of such myofibers.

On isolated cultivated myofibers, there was a reduced number of MuSC derived cells. EdU staining should be used to prove the suggested defect of proliferation in this assay. If EdU incorporation is not changed alternative explanations for the reduced number of MuSC descendants need to be addressed.

The authors superficially address the downstream mechanisms of Piezo1 function by staining for pMLC and by applying a Rho activator. pMLC staining needs to be normalized to Pax7 staining to be useful and different stages of MuSC differentiation need to be analysed for pMLC expression to exclude effects of cell number and differentiation status on pMLC.

In general, description of the statistical basis of the analysis and description of the statistical methods used is poor or not present, major improvements are needed to allow assessment of the data. The overall quality of the figures needs to be improved (e.g. consistency of style and size of the labelling).

The referees' critiques are italicized and our responses are printed in blue.

Referee #1:

We thank Reviewer 1 for carefully reading our manuscript and for the constructive comments. We added new data to address the concerns that were raised as described below.

Major issues:

1. Fig. 1A, the QSC should be called Freshly isolated SCs (FISCs) as it is an acceptable notion in the field of MuSC that FASC sorted cells are not real quiescent cells. The whole muscle tissue is used as a substitute for myofiber, which is not appropriate. The authors should examine Piezo1 expression profiles in the entire lineage development eg. QSC, FISC, ASC-24, ASC-48, ASC-72...DSC...confirm it is indeed highly induced in proliferating myoblasts but decreased in differentiating myofibers. This can also be done by analyzing the published RNA-seq data from various papers. In addition, in Fig 1A, it is known that the expression of Gapdh is induced during SC activation, so alternative controls such as 18S should be used to examine Piezo1 expression by RT-PCR. Moreover, quantitation of the semi-quantitative RT-PCR from at least 3 replicates should be performed.

We agree with this critique and have corrected the terms related to MuSCs. “QSC” was changed to “freshly isolated satellite cells (FISCs)”. We also performed RT-qPCR to examine the expression profiles of *Piezo1* during myogenesis and found that *Piezo1* is highly expressed in FISCs and proliferating MuSCs but is only marginally expressed in mature myofibers. As recommended by referee #1, *18S* expression was used as an alternative control for RT-qPCR. These results are shown in Fig. 1A.

We also performed an *in-silico* analysis based on the published RNA-seq data from two independent papers as shown in the figure below. In the left panel, expression profiles of QSCs and freshly-isolated MuSCs were analyzed and *Piezo1* expression was increased at 3 to 5 hours post-isolation. In the right panel, *Piezo1* is predominantly expressed in FISCs and the expression level is gradually decreased in ASCs and myofibers. These data further confirm that *Piezo1* is predominantly expressed in FISCs and the expression level of *Piezo1* is significantly decreased in myofibres.

Figure. *In silico* analysis of *Piezo1* expression

Machado et al. (PMID: 33609440) shows up-regulation of *Piezo1* expression in freshly isolated MuSCs (FISC) compared with that in quiescent MuSCs (QSC). Harada et al. (PMID: 29643389) shows down-regulation of *Piezo1* expression in activated MuSCs (1 day post-CTX injection) compared with that in FISCs.

2. Fig. 1B, Western blotting should be performed in samples from the above mentioned time points of SC lineage progression to confirm its expression dynamics. In addition, there is no description of Fig 1C or EV1B in the Result.

We conducted the suggested experiment to confirm the expression of PIEZO1 protein during myogenesis. PIEZO1 protein was clearly detected in cultured MuSCs isolated from *PIEZO1-tdTomato* mice, but not in mature skeletal muscle tissue. Although we could not detect it in FISCs because of the difficulty in preparing sample for western blot analysis, our data indicates that PIEZO1 is expressed in myogenic cells at the protein level. We have included this new data in Fig 1B.

3. I have major issues with Fig. 2, have the authors measured the deletion efficiency of *Piezo1* in SCs from the cKO mice? It should be evaluated on DNA, mRNA and protein levels. When were the SCs isolated after tamoxifen injection? In Fig 2A, what does each line of different color stand for? Also, the legend writing is not precise, 2Ca is not in the "upper panel" but "left panel".

We conducted RT-qPCR experiments based on referee #1's critique and found that *Piezo1* is quite efficiently deleted under our experimental conditions. These data were included in Figs 2D, 6B, and EV2E. We have included the schemes showing the time course of MuSCs isolation following tamoxifen injection in each figure.

We apologize for not providing a complete description in the figure legend. We have corrected this mistake in Fig 2A. We have also included the information in the legend as follows: "Representative traces of Ca^{2+} fluctuations in individual control MuSCs in the presence (2 mM Ca^{2+} ; left panel with light green, magenta, gray and dark blue traces) or absence (0 mM Ca^{2+} ; right panel with dark green, red, black and light blue traces) of extracellular Ca^{2+} ."

In EV2A, what do Piezo1Tm1a/Tm1c/Tm1d indicate? Why were 2 different chelators (EGTA and BAPTA-AM) were used? What is the difference between these two? In Figure 2G, representative images of the EdU staining should be shown.

We decided not to use the terms Tm1a, Tm1c, and Tm1d. Instead, we have used *Piezo1^{LacZ}*, *Piezo1^{flox}*, and *Piezo1⁻* in Fig EV2A.

We utilized two different chelators EGTA and BAPTA-AM for removal of extracellular and intracellular Ca^{2+} , respectively.

We agree with the concern related to Fig. 2G. Representative images of the EdU staining have been shown in Fig. 2H.

4. In Fig 2C, it is known that the use of Pax7CreERT2/+ causes heterologous deletion of the Pax7 gene, so Piezo1flox/flox; Pax7+/+ is not an appropriate control for the cKO mice; Instead, Pax7CreERT2/+ littermates should be used as controls.

We agree with the concern related to *Pax7^{CreERT2}* mice. As Referee #1 suggested, we performed control experiments using *Pax7^{CreERT2/+}* mice as true controls. These data are shown in Fig EV6.

5. The characterization of the mutant phenotype in Fig. 3 should be more completed for example by sampling muscles from different time points after CTX injection. The evaluation of fiber size should be done by showing the range of fiber size distribution instead of the average fiber size. Staining of eMyHC should also be performed to evaluate the degree of muscle regeneration.

We agree with the concerns related to Fig. 3 showing the histological abnormalities in *Piezo1* cKO muscle. As Referee #1 suggested, the range of fiber size distribution is shown in Fig 3 and EV3A – 3I. In addition, muscle samples from different time points (e.g., 14, 21, and 30 days) were analyzed. Although there was no statistical difference in fiber size, muscle weight tended to be reduced in *Piezo1* cKO. These data are included in Fig. EV3.

We also conducted embryonic MyHC staining, but the immunoreactivity was not clearly detected. Thus, we decided not to include the data in the manuscript.

Also in Fig EV4B, it looks like the fibers in Pax7CreERT2/+ are evidently smaller in size compared to the control, but the statistic result shows no difference.

The data in EV4B was re-evaluated based on the referee's concern. By showing the range of fiber size distribution, we confirmed that there was no statistical difference between wild-type and Pax7^{CreERT2/+} mice.

6. In Fig 6, Western blot should be performed to detect the change of pMLC after Piezo1 deletion. In Fig 6D, the authors should explain why CN03 treatment caused change of SC number in wide type group which was opposite to that of Piezo1 deletion group. Moreover, the author should prove whether CN03 treatment would rescue the abnormalities caused by Piezo1 deletion. Overall, I don't think the presented mechanism carries sufficient novelty considering very similar pathway has been shown in their previous paper using C2C12 myoblasts.

We strongly agree with the referee's critique regarding the detection of pMLC by western blot analysis. We performed the experiment; however, we could not clearly detect pMLC using our experimental conditions. Instead, we added new findings to reveal the downstream pathways of PIEZO1. As mentioned above, *Piezo1* deficiency causes a variety of abnormalities in MuSCs such as a reduction in the active form of Rho, abnormal MuSC activation (i.e., cell cycle progression) in the early phase, and chromosomal catastrophe during mitosis. Importantly, Rho activation by CN03 significantly rescued the phenotypes observed in *Piezo1*-deficient MuSCs, including reduction in the level of pMLC. These data are included in Figs 7G, 7H, 8G, and EV9F. We believe that these results are distinct from our previous paper showing the role of PIEZO1 in myotubes and provide insight into the mechanism underlying myogenesis.

7. The writing also needs to be polished. In Discussion, instead of interpreting their data, the authors write too much on mechanics sensing of MuSC and niche environment, which was not even studied in the Result. In a couple of places, I don't think the phrases used are precise. For example, on page 4 bottom, "However, the function of PIEZO1 in MuSC remains to be elucidated", this is not a precise justification for their study. On page 10, the use of "cell fate decision" is not precise in interpreting their related result.

We thank Referee #1 for the constructive comments on the Discussion section, which was revised based on the suggestions. Although we revealed that MuSC proliferation is affected by changes in substrate stiffness (as shown in EV8), we toned-down the description regarding mechanosensation associated with substrate stiffness.

We strongly agree with Referee #1's concern in the original manuscript: "*However, the function of PIEZO1 in MuSC remains to be elucidated, this is not a precise justification for their study*". In the revised manuscript, we present several new findings, such as the regulation of MuSC

activation and cell division. These data indicate that PIEZO1 plays multiple roles during myogenesis. Thus, we believe that the description is now suitable for describing the function of PIEZO1 in MuSCs.

We agree with Referee #1's concern related to "cell fate decision". We have changed this term in the revised manuscript.

Referee #2:

Overall, this study is interrogating an interesting biological process, it is well conducted and is using an appropriate mouse model to investigate the role of Piezo1 in vivo. However, some aspects of this work remain preliminary and have to be more rigorously addressed for publication.

We thank Referee #2 for carefully reading of our manuscript and for the constructive suggestions for improvement. We have addressed all of the points that were raised, in part, by generating new data as detailed below.

Specifically:

1. Based on published transcriptomic data, Piezo2 is also expressed in MuSCs. It has been shown that Piezo proteins can synergistically act in transducing mechanical signals and are involved in compensatory mechanisms. It would be interesting to complement Fig.1A by showing the impact of Piezo1 deletion on Piezo2 transcripts.

We agree with Referee #2's suggestion regarding PIEZO2 expression in *Piezo1*-deficient MuSCs. We evaluated the expression levels of *Piezo2* based on our RNA-seq data sets and found that quite low expression was evident in control and *Piezo1*-deficient MuSCs cultured in GM at day 3, as shown in the right panel. This indicates that PIEZO2 may not be involved in mechanosensation in MuSCs during myogenesis.

Figure: Expression of *Piezo2* mRNA in *Piezo1*-deficient MuSCs

2. Referring to Fig.1A, it is written "Piezo1 mRNA expression was clearly and moderately detected in QSCs and ASCs"; if anything, Fig.1A shows more Piezo1 in ASC. In any case, RT-qPCR, instead of semi-quantitative RT-PCR, should be performed to draw accurate quantitative conclusions.

We agree with Referee #2's constructive suggestion and conducted an RT-qPCR analysis on freshly isolated satellite cells (FISCs), cultured MuSCs, and skeletal muscle tissue. As shown in Fig. 1A, we confirmed the predominant expression of *Piezo1* mRNA in MuSCs and myoblasts, but only marginal expression in skeletal muscle.

2. A recent study (Eliazer et al. CSC 2019) demonstrated that a Wnt4-Rho-Rock axis is required for MuSC quiescence maintenance. This study also showed that genetic inhibition of Rho induced expansion of the MuSC pool (increase of Pax7+/BrdU+ cells) and MuSC activation (increase of Myod+). Given the link between Piezo1 and Rho, and the fact that Piezo1 is expressed in freshly isolated MuSCs (Fig. 1A), it would be interesting to investigate if Piezo1 cKO have an effect on the quiescent MuSC in resting muscle. To address this, the authors could score Pax7 cells (as well as for ectopic proliferation (EdU) and Myogenin cells) a few weeks (usually 3 to 4 weeks) after tamoxifen inductions.

As Referee #2 suggested, we evaluated the status of MuSCs in resting muscle at 1, 2, 4, and 16 weeks post-tamoxifen injection. The number of Pax7-positive MuSCs was clearly decreased at 2 and 16 weeks following TMX treatment, suggesting that PIEZO1 plays a role in quiescent MuSCs. Although this phenotype is very interesting, we decided not to include this data in the current version of this manuscript because of the space limitation.

[Figure removed by LSA Editorial Staff per Authors' request.]

3. *If no phenotype is found in quiescent Piezo1 cKO MuSCs, given the sensor property of Piezo1, it would be of great interest to challenge the cKO mice on a treadmill and see how the mutant MuSC respond to mechanical stimuli.*

As explained above, PIEZO1 plays a role in MuSCs under a resting state. Ideally, treadmill experiments would be useful to further investigate the function of PIEZO1. As it will take considerable time and the equipment would need to be purchased for a treadmill experiment, we will perform the suggested experiments in the future.

4. *Changes in cytoskeletal signaling and mechanical properties are early events during the transition from quiescence to activation. Consequently, it is difficult to estimate if the changes in pMLC are a consequence of activation or a direct consequence (direct target?) of Piezo1 deletion. Thus, the manuscript would be significantly improved by elucidating on the mechanism downstream of Piezo.*

-The authors should perform Western blot analysis of the levels of pMLC vs MLC in Control and Piezo cKO MuSCs, to show that the decrease of pMLC is not due to a decrease of total MLC protein.

-Does Rho activation rescue the phosphorylation of MLC in Piezo cKO MuSCs?

-What is the status of Rho mRNA and/or protein upon Piezo cKO? Is Rho less expressed or less functional (to phosphorylate pMLC)?

-Does Rho inhibition (EV.5E) impact pMLC?

As described in the response to Referee #1's critique, we strongly agree with the point related to the detection of pMLC by western blot analysis. However, we performed this experiment, but we could not clearly detect the band corresponding to pMLC under our experimental conditions.

Instead of western blot analysis of pMLC, we added new findings that reveal the downstream pathways associated with PIEZO1. As mentioned above, *Piezo1* deficiency causes a variety of abnormalities in MuSCs including a reduction in the active form of Rho, abnormal MuSC activation in the early phase, and chromosomal catastrophe during mitosis. Importantly, Rho activation by CN03 treatment partially restored the phenotypes observed in *Piezo1*-deficient MuSCs. These data are included in Figs. 7G, 7H, and 8G.

We agree with Referee #2's critique regarding pMLC levels following CN03 treatment. As shown in EV 9F, we have revealed that the Rho activator restores the phosphorylation of pMLC in *Piezo1* cKO.

We investigated Rho mRNA levels in *Piezo1* cKO and found no significant difference in *Rho A/B/C* mRNA expression between the control and *Piezo1* cKO.

Figure. *In silico* analysis of RhoA/B/C in Piezo1-cKO based on our RNA-seq data.

6. Fig.3B: Based on the images of this Figure, there seems to be a "homing" phenotype in the mutant, where the Pax7 cells are blocked in the interstitial space and not located under the basal lamina. It is not easy to estimate that based on a single picture, but I strongly recommend that the authors quantify this potentially interesting phenotype.

We strongly agree with Referee #2's suggestion. We investigated the homing phenotype in Piezo1 cKO based on images of Fig 3B and EV6E; however, we could not detect MuSCs that were blocked in the interstitial space and not located under the basal lamina. We will conduct the detailed analysis in the future study.

7. Fig.3D-E. The quantification does not correlate well with the pictures (or vice versa). It is also unclear how the authors define the "Area of regenerating fibers (graph axis label)".

Based on the Referee #2 suggestion, we included the exact data showing the changes in the cross-sectional area (CSA) in regenerating myofibers after CTX injection based on Laminin I/DAPI staining. These data are shown in Figs. 3G and 3H.

8. Please, provide statistics between datasets 1-3 (is the no. of Pax7 different in the cKO?) and 2-4 (is the Rho activator restoring the Pax7 cells?)

We apologize for the insufficient statistics in the original manuscript. We conducted a more robust statistical analysis, which is included in Fig 7H.

9. Fig.4D-F: The lower EdU observed in the mutant MuSC, is it due to a delay at the cell cycle entry? Given the proposed role of Piezo1 as a sensor of environmental stress, it would be relevant to assess if this gene has an impact on the activation time and entry to the cell cycle. This could be assessed by performing EdU chase experiments on freshly isolated MuSCs for 40-45h to capture the first S-phase.

We thank the Referee #2 for the constructive suggestions. We performed detailed EdU chase experiments. Surprisingly, as described above, our study revealed that *Piezo1* deficiency resulted in a variety of phenotypes, such as an increased number of EdU+ cells at the early phase (at 30 h post-isolation). We also confirmed this phenomenon by detection of Ki67-positive MuSCs. These results suggest that PIEZO1 has an effect on myogenesis including the regulation of cell cycle entry. These data are shown in Figures 4G-4H, 5F, EV5B, and EV6K-6L.

10. The authors have analyzed muscle sections on D7 post-Ctx injury and have concluded that "conditional deletion of Piezo1 in MuSCs delays myofiber regeneration after myofiber injury". It is strongly recommended that they also analyze the regeneration at a later time point, like D28 post-Ctx, in order to determine whether the repair defect upon Piezo1 ablation is maintained or restored by potential compensatory mechanisms.

We agree with the concerns with respect to Fig. 3 showing the histological abnormalities in *Piezo1* cKO muscle. As Referee #2 suggested, muscle samples from different time points (e.g., 14, 21, and 30 days) were analyzed. Although there was no statistical difference in the fiber size, muscle weight tended to be reduced in *Piezo1* cKO. These data are included in Fig. EV3.

11. Fig. 1C. Quantification is needed. Do all MuSC express PIEZO1-dtTom? In fact, on page 6, 2nd paragraph the authors refer to detection of Piezo1-dtTom based cytometry, however, the data are not presented. It recommended to include the FACS analysis with the corresponding quantifications in a figure.

We evaluated the expression of PIEZO1-tdTomato in isolated MuSCs and found that almost all Pax7-positive cells also expressed PIEZO1-tdTomato. This data is shown in Fig. 1E.

The MuSCs used in this study were isolated using PE-conjugated VCAM and APC-conjugated Sca1/CD31/CD45, as described in the Methods section. MuSCs from *Piezo1-tdTomato* mice were isolated as shown in the following figure.

Without PE-VCAM antibody

With PE-VCAM antibody

Figure. Representative FACS profiles for isolating MuSCs from *Piezo1-tdTomato* mice
Top panel: FACS profiles of mononuclear cells derived from limb muscles of *Piezo1-tdTomato* mice without anti-VCAM1-PE antibody. Bottom panel: FACS profiles of mononuclear cells derived from limb muscles of *Piezo1-tdTomato* mice by adding anti-VCAM1-PE antibody.

12. In Figure 2C the conditional *Piezo1* mice are introduced. Technically, this is not the appropriate control and it would be better to use *Pax7-CT2; Piezo+/+* treated with tamoxifen. Also, you should show efficiency of *Piezo1* deletion by RT-qPCR on FACS-isolated MuSCs.

We agree with the referee's concerns regarding the control mice and used *Pax7^{CreERT2}; Piezo1^{+/+}* mice as controls. These data are included in Fig. EV6A – 6M. Also, we evaluated the efficacy of *Piezo1* deletion by RT-qPCR on FACS-isolated MuSCs, which revealed that *Piezo1* gene expression was significantly reduced under these experimental conditions. This data is shown in Fig. 2D.

13. EV.1C: Piezo1-dtTom seems to also be expressed in Pax7-neg cells. What is the expression pattern of Piezo1 during myogenic commitment (in Myogenin+/Pax7- cells and freshly formed myotubes)?

As Referee #2 suggested, we measured the expression of PIEZO1-tdTomato in MuSCs and differentiated myotubes. Interestingly, PIEZO1 localization was clearly changed during differentiation. PIEZO1 was internalized in myogenin-positive cells and its expression level was clearly decreased in differentiated cells. These data are included in EV1C–1E. Although this phenotype is very interesting, we will examine the molecular mechanism underlying altered expression patterns in future studies.

Editorial comments:

i. The authors refer to freshly isolated MuSCs as quiescent, however, recent studies have shown that these cells are activated during the dissociation process. Please, change the designation from QSC to freshly-isolated or early-activated MuSCs.

We changed the terms related to MuSCs as suggested.

ii. Fig.1B, change labels from PIEZO1 to PIEZO1-dtTom. In the current format it is misleading.

We agree with this suggestion, and changed the labels from PIEZO1 to PIEZO1-tdT.

iii. Rephrase the title. Satellite cell function is too broad and vague description. It could be changed to "proliferative and regenerative function"?

We have performed a series of experiments based on the reviewers' suggestions and found that PIEZO1 plays multiple roles, such as maintaining the stemness at the early phase and completion of cell division at the late phase, indicating that PIEZO1 has a significant impact on MuSC function. Thus, we believe that the title of this manuscript: "The mechanosensitive Ca²⁺-permeable ion channel PIEZO1 promotes satellite cell function in skeletal muscle regeneration" is appropriate.

iv. *At the first sentence of the Introduction it is written "muscle regeneration after injuries induced by repeated contraction and relaxation of myofibers". Is this true? Injuries in general are linked to myofiber necrosis. Can the authors include references describing contraction/relaxation-induced regeneration and MuSC activation?*

We apologize for this incorrect description. We corrected the introduction paragraph in the revised manuscript.

v. *In Figure 2C the conditional Piezo1 mice are introduced. Please, consider the following recommendations: The mice indicated as "wild-type" are in fact Piezo1 flox/flox. Please, change to "Control".*

We agree with Referee #2's suggestion. The mice *Piezo1^{flox/flox}* are now indicated as the "control" in the revised manuscript.

vi. *EV1B is not cited in the main text.*

We thank Referee #2 for this suggestion. EV1B is now cited in the main text.

Referee #3:

The findings are interesting however interpretation of the results still is limited by some shortcomings. The paper focuses on the description of the MuSC-specific Piezo1 knock-out and might demonstrate the importance of Piezo1 for MuSC function, but the exact mechanisms of Piezo1 action in MuSCs remain largely unexplored.

We thank Reviewer 3 for carefully reading our manuscript and for the constructive suggestions. We have addressed almost all of the points that were raised, in part, by obtaining new data as detailed below:

Specific comments:

Quantification of peaks observed in WT MuSCs with and without calcium (Fig. 2B) seems not to match the representative traces depicted in Fig. 2A. The black trace in Fig. 2A has a peak amplitude of > 0.1, however there is no such point in the quantification in Fig. 2B. The description of how the quantification is performed is poor. There should be a clear description of how this quantification was performed (e.g. highest peak per xx sec trace).

We apologize for the insufficient description regarding the quantification of Ca²⁺ measurements. Amplitudes of Ca²⁺ fluctuations were calculated based on the following equation:

$$(\text{Amplitude}) = ((\text{maximum value of the F340/F380 ratio}) - (\text{minimum value of the F340/F380 ratio})) / 2$$

In the figure legend, there should be a description of how many individual cell isolated from how many animals were observed, and it should be made clear how peaks were identified and how the amplitude of the peaks was determined. Based on this, the statistical test used for the quantified data needs to be carefully considered. This criticism similarly applies to Fig. 2D-E. The statement in the figure legend "data represent means +/- SEM" is wrong, in the figure 2 B/E there are no errorbars but datapoints plus a bar indicating means or median. There corresponding author should consult the experimenters for a better description of the data.

We apologize for the insufficient description regarding the experimental conditions. Detailed information (e.g., the number of mice, individual cells) are included in the figure legends.

EdU positive MuSCs were counted after application of BAPTA-AM in vitro. Representative pictures of those staining should be included into the manuscript and the absolute numbers or percentage of EdU incorporating cells should be included to allow judgment of the proliferative activity of those cells in culture. The authors need to indicate the number of animals used for isolating MuSCs, how many cells were counted per animal, or how many independent experiments were performed in the figure legend to reveal the statistical basis of the bar graphs.

We agree Referee #3's suggestions. Representative images and the percentage of EdU-positive MuSCs are included in Figs. 2H – 2I. Detailed information including the numbers of mice and cells are included in the figure legends.

To confirm the CTX injury the authors should add pictures to figure 3 that clearly demonstrate centrally localized nuclei in the regenerating muscle tissue. The statement on infiltrating immune cells in the results section needs to be qualified by appropriate IF staining or by any other means. Again, the statistical basis of the bar graphs need to be clarified. The authors should carefully discuss all possible explanations for the reduced regeneration of myofibers after CTX injection.

We agree with Referee #3's suggestions. We added the images showing centrally localized nuclei in the regenerating tissue. These images are shown in Fig 3G.

Regarding the investigation of immune cells, we decided not to include the description as our focus on this revised manuscript is the role of PIEZO1 in MuSC activation and proliferation, rather than the phenotype of *Piezo1*-deficient mice. Thus, Fig 3G, which contained a description of infiltrating immune cells, has been deleted.

The number of Pax7 positive MuSCs was quantified in muscle sections and seems to be reduced in Piezo1 KO mice. The authors also should use and quantify additional markers of quiescent and proliferating MuSCs in combination with Pax7 staining to demonstrate the physiological state of the MuSCs in vivo.

We agree with Referee #3's suggestion. To further investigate the role of PIEZO1, we utilized another mouse line, cKO YFP (*Piezo1^{flox/flox} Pax7^{CreERT2/+} Rosa26^{YFP/+}*), that expresses YFP in MuSCs and an MuSC-derived lineage. Our results demonstrate that the number of YFP-expressing cells was reduced at 7 days post-CTX injection, as shown in EV6D – 6F. We have used additional markers to quantify quiescent and proliferating MuSCs. As shown in Figs. 4E and 4F, EdU+ and M-cadherin+ (a marker for quiescent and proliferating MuSCs) cells are significantly reduced in injured muscle. These data further confirm our results in Figs 4A – 4C using an anti-Pax7 antibody.

In addition, the authors need to include Pax7-CreERT pos controls to Fig. 4A-C. The number of positive cells should be indicated per μm^2 .

We understand the importance of the control experiments for this *Pax7^{CreERT2}* line. We added the data in Figs. EV7D–7F to eliminate the possibility that phenotypes observed in *Piezo1*-deficient mice are the result of heterologous insertion of Cre into the *Pax7* locus. As suggested, the numbers of positive cells are indicated per μm^2

The type of muscle used for Fig. 4A-C should be indicated in the figure legend and at least one different muscle should be included in this analysis for a more general statement on the MuSC population of the skeletal muscle.

We agree with Referee #3's suggestion. The type of muscle used for Figs 4A – 4C was indicated as “TA” in the figure legend. We have evaluated the phenotypes in other tissues, including gastrocnemius tissues, and found that a reduction in tissue weight of gastrocnemius muscle was evident, similar to that of TA muscle. This data is included in EV3D. We also conducted a histological analysis on gastrocnemius muscle showing histological abnormalities, as described below. In addition, we have also analyzed the proliferating capacity of MuSCs in extensor digitorum longus (EDL) muscle, as shown in Fig 5G and 5H, EV5C – 5F, and EV6M.

Figure. Histological analysis of gastrocnemius muscle in *Piezo1* cKO mice one week after injury.

Left panel: control muscle. Right panel: *Piezo1* cKO muscle. Scale bar 100 μ m

EdU incorporation into MuSCs was quantified after CTX injury; representative pictures of sections used for quantification should be included into the figure.

We included the representative images of EdU-incorporated muscles in Fig 4E.

There is massive, non-physiological stimulation of proliferation after CTX injury. To explore why there are less MuSCs in resting muscle, data on EdU incorporation in undisturbed muscle should be included into the analysis.

We agree with Referee #3's suggestion. To evaluate MuSC status in detail, we performed an immunofluorescent analysis of Ki67 (a proliferation marker) in MuSCs of intact muscle at 1-week post-TMX administration. Interestingly, as shown in the figure below, *Piezo1*-deficient MuSCs had an increased number of Pax7-negative, Ki67-positive cells ($P < 0.01$, compared with its control). Since the number of Pax7-positive MuSCs in *Piezo1* cKO was comparable to that in control at 0 day post-TMX treatment (see Fig 5A, 5B; EV6A-6C, EV6H and 6I), these results indicate that *Piezo1*-deficient MuSCs are prone to exit quiescence and enter the cell cycle.

[Figure removed by LSA Editorial Staff per Authors' request.]

The number of Pax7 positive cells on isolated myofibers is not reduced. On the other hand, there is a reduced number of MuSCs / view field in muscle sections. This is an obvious contradiction and the authors need to explain this. One possible explanation would be a different size/number of myofibers. The number of Pax7 positive cells per fiber appears quite high; the authors should include representative pictures of such myofibers.

For detection of Pax7-positive cells, we analyzed isolated myofibers without CTX injection (Fig 5B) and muscle sections at 7 days post-CTX injection (Fig 4B). Thus, it is possible that the contradiction suggested by Referee #3 may be the result of differences in experimental conditions.

On isolated cultivated myofibers, there was a reduced number of MuSC derived cells. EdU staining should be used to prove the suggested defect of proliferation in this assay. If EdU incorporation is not changed alternative explanations for the reduced number of MuSC descendants need to be addressed.

We agree with Referee #3's critiques. In the revised manuscript, we conducted EdU incorporation assays on isolated cultivated myofibers. Surprisingly, our study revealed that *Piezo1* deficiency resulted in an increased percentage of EdU-positive MuSCs at the early phase (at 30 h post-isolation). However, the number of MuSCs or MuSCs-derived cells decreased at the later phase (at 48 and 72 h post-isolation). These results suggest that PIEZO1 has multiple effects on myogenesis including regulation of the cell cycle entry and promotion of cell proliferation. These data are shown in Figures 5F – 5H, EV5C – 5F, and EV6K–6M.

The authors superficially address the downstream mechanisms of Piezo1 function by staining for pMLC and by applying a Rho activator. pMLC staining needs to be normalized to Pax7 staining to be useful and different stages of MuSC differentiation need to be analysed for pMLC expression to exclude effects of cell number and differentiation status on pMLC.

In general, description of the statistical basis of the analysis and description of the statistical methods used is poor or not present, major improvements are needed to allow assessment of the data. The overall quality of the figures needs to be improved (e.g. consistency of style and size of the labelling).

We agree with Referee #3's suggestion. We have re-evaluated pMLC expression based on Pax7 intensity. Although pMLC intensity was significantly decreased in *Piezo1* cKO compared with

control cells, there was no statistical significance in the pMLC/PAX7 ratio, as shown below. One possible explanation is that *Piezo1*-deficiency affects MuSC status, thus the expression of Pax7 could be altered in a subpopulation of MuSCs.

[Figure removed by LSA Editorial Staff per Authors' request.]

Although we could not evaluate pMLC expression by calculating the pMLC/Pax7 ratio, we showed that PIEZO1 is required for generation of the active form of Rho. These data are shown in Figs 7A–7C. Moreover, the RhoA activator, CN03, clearly restored the phenotypes: (i) abnormal cell cycle entry at the early phase, (ii) reduced proliferation of MuSCs at the late phase, and (iii) decreased phosphorylation level of MLC, all of which were observed in *Piezo1* cKO MuSCs. These data are included in Fig 7F – 7H, EV9C and 9F. Our findings listed above further strengthen the significance of the PIEZO1 – Rho – pMLC axis in MuSCs.

Referee #1 Review

Report for Author:

In the revised version of the manuscript titled "The mechanosensitive Ca²⁺-permeable ion channel PIEZO1 promotes satellite cell function in skeletal muscle regeneration", Kotaro Hirano et. al. has largely addressed my concerns raised in the first round of revision but there are still some issues to be addressed (see below). Unfortunately, a very recent article published by Peng Y. et.al. (1) reports very similar findings, which affects the novelty and significance of this study. Both studies demonstrate PIEZO1 function in muscle stem cells and muscle regeneration using the muscle stem cell specific PIEZO1 inducible knockout mice. Although the underlying mechanisms are somewhat different, many findings are similar, for example, both show the expression dynamics of PIEZO1 in muscle stem cells, and its impact on muscle stem cell function and muscle repair. The study by Peng Y. et.al. in fact presents more comprehensive and solid evidence supporting PIEZO1 function in muscle stem cells and muscle repair.

1. In Figure 2, although the PIEZO1 RNA expression has been mentioned in PIEZO1 cKO mice, the protein expression of PIEZO1 still needs to be examined to confirm the deletion efficiency.
2. In Figure 4E, Pax7 and EdU double staining should be performed to measure the changed MuSC activation and proliferation in vivo in PIEZO1 cKO mice. Besides, in Figure 4G, since a large portion of MuSCs should enter cell cycle at 36h after seeding, it is surprising to see only around 15% EdU+ MuSCs present after cultured for 40h.
3. In Figure 6, the authors should explain why the RNA-seq was done in MuSCs cultured for 3 days. And in Figure 6C, the number of up-regulated and down-regulated genes should be indicated.
4. In Figure 7, the author mentioned that "although the active form of Rho was expected to be up-regulated based on our RNA-seq analysis", the RNA expression change of Rho determined by RNA-seq should be shown in Ctrl/ PIEZO1 cKO mice.
5. There are still a lot of language problems that need to be carefully edited. For examples, in the abstract, "This results" should be changed to "These results".

1. Peng, Y., Du, J., Günther, S., Guo, X., Wang, S., Schneider, A., ... & Braun, T. (2022). Mechano-signaling via Piezo1 prevents activation and p53-mediated senescence of muscle stem cells. *Redox biology*, 52, 102309.

Referee #3 Review

Report for Author:

The authors address most of the reviewers questions in an adequate manner, although some points (see below) still need to be changed. Nevertheless, the manuscript might be an interesting resource for future understanding of the function of Piezo1 in muscle regeneration

The statement "Data represents means \pm SEM for figure 2B/2F" still does not reflect what is depicted in the figure. In response to referee comment the authors describe removal of "Fig 3G, which contained a description of infiltrating immune cells". Former Figure 3G contained a quantification of collagen I deposition and is still present as Fig 3F.

The author's state to have added data on Pax7-CreERT pos controls to EV7D-7F. Figure EV7 does not contain D-F. I assume the authors refer to Fig EV6??

October 24, 2022

Re: Life Science Alliance manuscript #LSA-2022-01783-T

Dr. Yuji Hara
School of Pharmaceutical Sciences, University of Shizuoka,
Department of Integrative physiology
52-1 Yada
Suruga-ku
Shizuoka, Shizuoka 422-8526
Japan

Dear Dr. Hara,

Thank you for submitting your manuscript entitled "The mechanosensitive Ca²⁺-permeable ion channel PIEZO1 promotes satellite cell function in skeletal muscle regeneration" to Life Science Alliance. We invite you to re-submit the manuscript, revised to address the Reviewers' remaining comments.

Thank you for this interesting contribution to Life Science Alliance. We are looking forward to receiving your revised manuscript.

Sincerely,

B. MANUSCRIPT ORGANIZATION AND FORMATTING:

Responses to the editors' and referees' critiques:

Please note that *original referees' comments are in Italicized print*, and that **our response is printed in blue**. We have carefully edited our manuscript as shown in red characters in the related manuscript file.

Referee #1:

In the revised version of the manuscript titled "The mechanosensitive Ca²⁺-permeable ion channel PIEZO1 promotes satellite cell function in skeletal muscle regeneration", Kotaro Hirano et. al. has largely addressed my concerns raised in the first round of revision but there are still some issues to be addressed (see below). Unfortunately, a very recent article published by Peng Y. et.al. (1) reports very similar findings, which affects the novelty and significance of this study. Both studies demonstrate PIEZO1 function in muscle stem cells and muscle regeneration using the muscle stem cell specific PIEZO1 inducible knockout mice. Although the underlying mechanisms are somewhat different, many findings are similar, for example, both show the expression dynamics of PIEZO1 in muscle stem cells, and its impact on muscle stem cell function and muscle repair. The study by Peng Y. et.al. in fact presents more comprehensive and solid evidence supporting PIEZO1 function in muscle stem cells and muscle repair.

We would like to thank reviewer #1 for the careful reading and constructive comments on our manuscript. Although reviewer #1 stated a concern raised by a recent paper (Peng et al., *Redox Biology*, 2022) published during our review process, we believe that our manuscript provides marked impact that will be appreciated by the broad readership of *Life Science Alliance*, owing to the following reasons:

1) Role of PIEZO1 as a 'mechanosensitive ion channel':

Since the early 1990s, there has been an urgent demand to understand the mechanism underlying the biophysical force-mediated regulation of muscle satellite cells (MuSCs). In our manuscript, we provide the first concrete evidence showing that PIEZO1 acts as a Ca²⁺-permeable mechanosensor that is activated by changes in substrate stiffness surrounding MuSCs and is critical for the completion of myogenesis (see Fig 2 and Fig S8 in our manuscript).

2) PIEZO1-mediated signaling pathways in myogenesis:

In this manuscript, we identified Rho-MLC as a PIEZO1-mediated intracellular signaling pathway and revealed that the PIEZO1-Rho axis is essential not only for the proliferation of MuSCs but also for the maintenance of MuSC stemness.

3) Elucidation of the novel function of PIEZO1:

One of the important findings of our manuscript is defining the localization of PIEZO1 during myogenesis. Our genetic approach revealed that localisation of the mechanosensor PIEZO1 is dramatically changed throughout myogenesis; a specific example of this is the accumulation of PIEZO1 at the midbody of MuSCs. Cell division consists of multiple processes in which a cell undergoes major structural and mechanical deformation. However, it is still unclear how mechanosensors and mechanical signals contribute to the progression of cell division. Our results provide concrete evidence showing that PIEZO1 plays a pivotal role in the division of MuSCs, which ultimately determines their cell fate.

Taking all of these points into consideration, our findings describe biologically novel mechanisms and will be of interest to audiences in a broad range of disciplines, not only skeletal muscle biologists, but also cell biologists and medical scientists. We hope referee #1 will now share our enthusiasm for this revised manuscript and will now recommend publication in *Life Science Alliance*.

1. In Figure 2, although the PIEZO1 RNA expression has been mentioned in PIEZO1 cKO mice, the protein expression of PIEZO1 still needs to be examined to confirm the deletion efficiency.

Based on referee #1's suggestion, we examined the protein expression of PIEZO1 through immunofluorescent analysis using an anti-PIEZO1 antibody (NBP1-78446, Novus Biologicals), as Peng et al., used in their study. However, we observed an immunofluorescent signal at the plasma membrane of MuSCs in *Piezo1*-deficient mice and the controls. This result suggested that the antibody may not specifically detect the native PIEZO1 protein in MuSCs (refer Figure below). In addition to immunofluorescence analyses, we measured Ca^{2+} in isolated MuSCs using the PIEZO1-specific agonist Yoda1 and found that increase in Ca^{2+} was clearly blunted in *Piezo1*-deficient MuSCs (refer Figures S2B, S2C, S2D, S2F, and S2G in our manuscript). These results further confirmed that the *Piezo1* gene was efficiently deleted in *Piezo1*-cKO MuSCs.

Figure. Detection of PIEZO1 with anti-PIEZO1 antibody in MuSCs

Detection of Pax7 and PIEZO1 in MuSCs on freshly isolated myofibres and myofibres cultured for 1 to 3 days in plating medium. Left: control, Right: *Piezo1* cKO. Scale bar: 10 μ m.

2. In Figure 4E, Pax7 and EdU double staining should be performed to measure the changed MuSC activation and proliferation *in vivo* in *PIEZO1* cKO mice.

We performed the suggested experiments to evaluate changes in MuSC activation and proliferation *in vivo*. Our results revealed that the number of EdU / Pax7 double positive cells clearly increased in *Piezo1*-cKO MuSCs 30 h post-CTX injection (Fig A – C), but significantly decreased at 3 days post CTX injection (Fig D – F). These results indicated that PIEZO1 played multiple roles in regulating MuSC activation and promoting MuSC proliferation *in vivo*. This was consistent with our *ex vivo* studies (Fig 4G – J, Fig 5F – H, Fig S5A – S5F, Fig S6K – S6M in our manuscript). However, for reason of space we have chosen not to include this data in the manuscript.

Figure. *In vivo* EdU incorporation assay on Pax7 positive MuSCs during muscle regeneration

A – C: EdU incorporation assay on Pax7 positive MuSCs to evaluate MuSC activation *in vivo*.

A. The timecourse for induction of *Piezo1* deficiency, injection of cardiotoxin and EdU, and isolation of regenerated muscle samples. **B.** Detection of Pax7-positive and EdU-incorporated MuSCs (arrowheads) in cross-sections obtained from control (upper panels) and *Piezo1* cKO muscle (lower panels) at 30 h post cardiotoxin injury. **C.** Quantification of the ratio of EdU-positive cells per Pax7-positive MuSC (N = 3 mice per condition). Bar graphs represent mean ± S.E.M, * P < 0.05. Scale bar: 100 μm.

D – F: EdU incorporation assay on Pax7 positive MuSCs for evaluating MuSCs proliferation *in vivo*.

D. The timecourse for induction of *Piezo1* deficiency, injection of cardiotoxin and EdU, and

isolation of regenerated muscle samples. **E.** Detection of Pax7-positive and EdU-incorporated MuSCs (arrowheads) in cross-sections obtained from control (upper row) and *Piezo1* cKO muscle (lower row), 3 days post cardiotoxin injury. **F.** Quantification of the ratio of EdU-positive cells per Pax7-positive MuSCs (N = 3 mice for control; N=5 mice for *Piezo1* cKO per condition). Bar graphs represent mean + S.E.M. * P < 0.05. Scale bar: 100 μ m.

Besides, in Figure 4G, since a large portion of MuSCs should enter cell cycle at 36h after seeding, it is surprising to see only around 15% EdU+ MuSCs present after cultured for 40h.

As indicated by referee #1, previous investigators (Machado et al., 2021) reported that approximately 50% of MuSCs entered the cell cycle 40 h after seeding. In contrast, Rodgers et al., showed that approximately 20% of the isolated quiescent MuSCs were EdU-positive at 40 h after seeding, which corresponded with our data. These results imply that the speed of cell cycle entry may depend on the culture conditions provided at each laboratory.

In addition, we examined the timecourse of EdU incorporation into isolated MuSCs to further validate our conditions of culturing. As reported by Yamaguchi et al. (2015), although only 1% of MuSCs were EdU-positive 24 h after plating, a majority of MuSCs enter the cell cycle 72 h after plating (see Fig A – C below). These results confirmed that MuSCs were activated under our experimental conditions.

References: Machado L et al., *Cell Stem Cell*, 2021 (PMID: 33609440); Rodgers JT et al., *Nature*, 2014 (PMID: 24870234); Yamaguchi M et al., *Cell Reports*, 2015 (PMID: 26440893).

Figure. *Ex vivo* incorporation of EdU into isolated MuSCs

A. The timecourse of MuSC culture in the growth medium in the presence of EdU (10 μ M). MuSCs were fixed at the indicated time points.

B – C: *Ex vivo* incorporation of EdU into isolated MuSCs. **B.** Representative images of fixed MuSCs at the indicated time points. **C.** Quantification of the ratio of EdU-positive MuSCs (N = 3 mice per condition). Bar graphs represent mean + S.E.M. Scale bar: 100 μ m.

3. In Figure 6, the authors should explain why the RNA-seq was done in MuSCs cultured for 3 days. And in Figure 6C, the number of up-regulated and down-regulated genes should be indicated.

We acknowledge and understand Referee #1's concern regarding our experimental design for RNA-seq analysis. We analysed 3-day-cultured MuSCs in the RNA-seq study to elucidate the role of PIEZO1 during MuSC proliferation. As our manuscript revealed that PIEZO1 played multiple roles during myogenesis, it would be interesting to perform RNA-seq on *Piezo1*-cKO MuSCs at different time points in future studies.

The number of up-regulated and down-regulated genes is indicated in the legend of Figure 6C.

4. In Figure 7, the author mentioned that "although the active form of Rho was expected to be up-regulated based on our RNA-seq analysis", the RNA expression change of Rho determined by

RNA-seq should be shown in Ctrl/ PIEZO1 cKO mice.

We investigated *Rho* mRNA levels in *Piezo1* cKO, but no significant difference in *Rho A/B/C* mRNA expression was observed between the control and *Piezo1* cKO (see Figure below). For reason of space we have chosen not to include this data in the manuscript.

Figure. *In silico* analysis of *RhoA/B/C* in *Piezo1-cKO* based on our RNA-seq data.

5. There are still a lot of language problems that need to be carefully edited. For examples, in the abstract, "This results" should be changed to "These results".

We thank Referee #1 for valuable comments. We have carefully edited our manuscript as shown in red characters in the related manuscript file.

Referee #3:

The authors address most of the reviewers questions in an adequate manner, although some points (see below) still need to be changed. Nevertheless, the manuscript might be an interesting resource for future understanding of the function of Piezo1 in muscle regeneration.

We thank referee #3 for the careful consideration of our manuscript and finding our study interesting. We have made the appropriate changes as suggested by referee #3.

The statement "Data represents means \pm SEM for figure 2B/2F" still does not reflect what is depicted in the figure.

We apologise for the errors in data representation. We have addressed this by deleting the means \pm SEM, from the figure legends.

In response to referee comment the authors describe removal of "Fig 3G, which contained a description of infiltrating immune cells". Former Figure 3G contained a quantification of collagen I deposition and is still present as Fig 3F.

We acknowledge and apologise for the incorrect response to the Referee's comment regarding Fig 3G. The former Fig 3G, contained a description regarding collagen I deposition, which was not related to infiltrating immune cells. Therefore, we have deleted this description from the manuscript.

The author's state to have added data on Pax7-CreERT pos controls to EV7D-7F. Figure EV7 does not contain D-F. I assume the authors refer to Fig EV6??

We apologize for this mistake. We refer the added data as Fig S6D – S6F.

November 3, 2022

RE: Life Science Alliance Manuscript #LSA-2022-01783-TR

Dr. Yuji Hara
University of Shizuoka
School of Pharmaceutical Sciences, Department of Integrative physiology
52-1 Yada
Suruga-ku
Shizuoka, Shizuoka 422-8526
Japan

Dear Dr. Hara,

Thank you for submitting your revised manuscript entitled "The mechanosensitive ion channel PIEZO1 promotes satellite cell function in muscle regeneration". We would be happy to publish your paper in Life Science Alliance pending final revisions necessary to meet our formatting guidelines.

- please add ORCID ID for corresponding author--you should have received instructions on how to do so
- please add a Running Title in our system
- please upload all figure files as individual ones, including the supplementary figure files; all figure legends should only appear in the main manuscript file
- please upload your main manuscript text as an editable doc file
- please add the Twitter handle of your host institute/organization as well as your own or/and one of the authors in our system

A. FINAL FILES:

B. MANUSCRIPT ORGANIZATION AND FORMATTING:

Sincerely,

November 8, 2022

RE: Life Science Alliance Manuscript #LSA-2022-01783-TRR

Dr. Yuji Hara
University of Shizuoka
School of Pharmaceutical Sciences, Department of Integrative physiology
52-1 Yada
Suruga-ku
Shizuoka, Shizuoka 422-8526
Japan

Dear Dr. Hara,

Thank you for submitting your Research Article entitled "The mechanosensitive ion channel PIEZO1 promotes satellite cell function in muscle regeneration". It is a pleasure to let you know that your manuscript is now accepted for publication in Life Science Alliance. Congratulations on this interesting work.

DISTRIBUTION OF MATERIALS:

Again, congratulations on a very nice paper. I hope you found the review process to be constructive and are pleased with how the manuscript was handled editorially. We look forward to future exciting submissions from your lab.

Sincerely,
